# MoMo: Momentum Models for Adaptive Learning Rates

## Abstract

Training a modern machine learning architecture on a new task requires extensive learning-rate tuning, which comes at a high computational cost. Here we develop new adaptive learning rates that can be used with any momentum method, and require less tuning to perform well. We first develop MoMo, a **Mo**mentum **Mo**del based adaptive learning rate for `SGD-M` (Stochastic gradient descent with momentum). `MoMo` uses momentum estimates of the batch losses and gradients sampled at each iteration to build a model of the loss function. Our model also makes use of any known lower bound of the loss function by using *truncation*, e.g. most losses are lower-bounded by zero. We then approximately minimize this model at each iteration to compute the next step. We show how `MoMo` can be used in combination with any momentum-based method, and showcase this by developing `MoMo-Adam` - which is `Adam` with our new model-based adaptive learning rate. Additionally, for losses with unknown lower bounds, we develop on-the-fly estimates of a lower bound, that are incorporated in our model. Through extensive numerical experiments, we demonstrate that `MoMo` and `MoMo-Adam` improve over `SGD-M` and `Adam` in terms of accuracy *and* robustness to hyperparameter tuning for training image classifiers on `MNIST`, `CIFAR10`, `CIFAR100`, `Imagenet`, recommender systems on the `Criteo` dataset, and a transformer model on the translation task `IWSLT14`.

## 1 Introduction

Training of a modern production-grade large neural network can cost over 1 million dollars in compute. For instance, the cost for the *Text-to-Text Transfer Transformer* T5-model (Raffel et al., 2020) is estimated to be more than 1.3 million dollars for a single run (Sharir et al., 2020). What makes training models so expensive is that multiple runs are needed to tune the hyperparameters, with arguably the most important parameter being the learning rate. Indeed, finding a good learning-rate schedule plays a disproportionately large role in the resulting test error of the model, with one extensive study showing that it was at least as important as the choice of optimizer (Schmidt et al., 2021).

Here, we develop adaptive learning rates that can be used together with any momentum-based method. To showcase our method, we apply our learning rates to `SGD-M` (Stochastic Gradient Descent with momentum) and to `Adam` (Kingma & Ba, 2015), which gives the `MoMo` and `MoMo-Adam` method, respectively. We make use of model-based stochastic optimization (Asi & Duchi, 2019; Davis & Drusvyatskiy, 2019; Chadha et al., 2021), and leverage that loss functions are bounded below (typically by zero) to derive our new MoMo (**Mo**del-based **Mo**mentum) adaptive learning rate.

### 1.1 The Model-Based Approach

Consider the problem

$$\min_{x \in \mathbb{R}^d} f(x), \quad f(x) := \mathbb{E}_{s \sim \mathcal{D}}\left[f(x, s)\right], \tag{1}$$

where $f(x, s)$ is a loss function, $s$ is an input (mini-batch of data), and $x$ are the parameters of a model we are trying to fit to the data. We assume throughout that $f(x, s) \geq 0$, which is the

case for most loss functions[1]. We also assume that $f(\cdot, s)$ is continuously differentiable for all $s \in \mathcal{D}$, that there exists a solution $x^*$ to (1) and denote the optimal value by $f^* := f(x^*) \in \mathbb{R}$.

In our main algorithms `MoMo` and `MoMo-Adam` (Algorithms 1 and 2), we present adaptive learning rates[2] for `SGD-M` and `Adam`, respectively. To derive `MoMo` and `MoMo-Adam`, we use the model-based viewpoint, which is often motivated by the Stochastic Proximal Point (`SPP`) (Asi & Duchi, 2019; Davis & Drusvyatskiy, 2019) method. At each iteration, `SPP` samples $s_k \sim \mathcal{D}$, then trades-off minimizing $f(x, s_k)$ with not moving too far from the current iterate $x^k$. Given a learning rate $\alpha_k > 0$, this can be written as

$$x^{k+1} = \underset{x \in \mathbb{R}^d}{\arg\min} \, f(x, s_k) + \tfrac{1}{2\alpha_k} \left\| x - x^k \right\|^2 . \tag{2}$$

Since this problem needs to be solved at every iteration, it needs to be fast to compute. However, in general (2) is difficult to solve because $f(x, s_k)$ can be a highly nonlinear function. Model-based methods replace $f(x, s_k)$ by a simple model $m_k(x)$ of the function (Asi & Duchi, 2019; Davis & Drusvyatskiy, 2019), and update according to

$$x^{k+1} = \underset{x \in \mathbb{R}^d}{\arg\min} \, m_k(x) + \tfrac{1}{2\alpha_k} \left\| x - x^k \right\|^2 . \tag{3}$$

`SGD` can be formulated as a model-based method by choosing the model to be the linearization of $f(x, s_k)$ around $x^k$, that is

$$m_k(x) = f(x^k, s_k) + \left\langle \nabla f(x^k, s_k), x - x^k \right\rangle . \tag{4}$$

Using the above $m_k(x)$ in (3) gives the `SGD` update $x^{k+1} = x^k - \alpha_k \nabla f(x^k, s_k)$, see (Robbins & Monro, 1951; Asi & Duchi, 2019).

Our main insight for developing the `MoMo` methods is that we should build a model directly for $f(x)$, and not $f(x, s_k)$, since our objective is to minimize $f(x)$. To this end, we develop a model $m_k(x)$ that is a good approximation of $f(x)$ when $x$ is close to $x^k$, and such that (3) has a simple closed form solution. Our model uses momentum estimates of past gradients and loss values to build a model $f(x)$. Finally, since the loss function is positive, we also impose that our model be positive.

## 1.2 Background and Contributions

**Momentum and model-based methods.** The update formula of many stochastic methods such as `SGD` can be interpreted by taking a proximal step with respect to a model of the objective function (Asi & Duchi, 2019; Davis & Drusvyatskiy, 2019). Independently of this, (heavy-ball) momentum (Polyak, 1964; Sebbouh et al., 2021) is incorporated into many methods in order to boost performance.

*Contributions.* Here we give a new model-based interpretation of momentum, namely that it can be motivated as a model of the objective function $f(x)$ by averaging sampled loss functions. This allows us to naturally combine momentum with other model-based techniques.

**Lower bounds and truncated models.** One of the main advantages of the model-based viewpoint (Asi & Duchi, 2019; Davis & Drusvyatskiy, 2019) is that it illustrates how to use knowledge of a lower bound of the function via truncation. Methods using this truncated model are often easier to tune (Meng & Gower, 2023; Schaipp et al., 2023).

*Contributions.* By combining the model-based viewpoint of momentum with a truncated model we arrive at our new `MoMo` method. Since we are interested in loss functions, we can use zero as a lower bound estimate in many learning tasks. However, for some tasks such as training transformers, the minimal loss is often non-zero. If the non-zero lower bound is known, we can straightforwardly incorporate it into our model. For unknown lower bound

---

[1]We choose zero as a lower bound for simplicity, but any constant lower bound could be handled.

[2]Here the term *adaptivity* refers to a scalar learning rate that changes from one iteration to the next by using easy-to-compute quantities. This is different from the notion of adaptivity used for `Adam` or `AdaGrad` (Duchi et al., 2011), where the learning rate is different for each coordinate. We refer to the latter meaning of adaptivity as *preconditioning*.

values we also develop new online estimates of a lower bound in Section 4. Our estimates can be applied to any stochastic momentum-based method, and thus may be of independent interest. Our main influence for this development was D-adaptation (Defazio & Mishchenko, 2023) which develops an online estimate of the distance to the solution.

**Adaptive methods.** In practice, tuning learning-rate schedules is intricate and computationally expensive. `Adam` (Kingma & Ba, 2015) and variants such as `AdamW` (Loshchilov & Hutter, 2019), are often easier to tune and are now being used routinely to train DNNs across a variety of tasks. This and the success of `Adam` have incentivised the development of many new adaptive learning rates, including approaches based on coin-betting (Orabona & Tommasi, 2017), variants of `AdaGrad` (Duchi et al., 2011; Defazio & Mishchenko, 2023), and stochastic line search (Vaswani et al., 2019). Recent work also combines parameter-free coin betting methods with truncated models (Chen et al., 2022).

*Contributions.* Our new adaptive learning rate can be combined with any momentum based method, and even allows for a preconditioner to be used. For example, `Adam` is a momentum method that makes use of a preconditioner. By using this viewpoint, together with a lower bound, we derive `MoMo-Adam`, a variant of `Adam` that uses our adaptive learning rates.

**Adaptive Polyak step sizes.** For convex, non-smooth optimization, Polyak proposed an adaptive step size using the current objective function value $f(x^k)$ and the optimal value $f^*$ (Polyak, 1987). Recently, the Polyak step size has been adapted to the stochastic setting (Berrada et al., 2020; Gower et al., 2021; Loizou et al., 2021; Orvieto et al., 2022). For example, (Loizou et al., 2021) proposed

$$x^{k+1} = x^k - \min\left\{\gamma_b, \frac{f(x^k, s_k) - \inf_z f(z, s_k)}{c\|\nabla f(x^k, s_k)\|^2}\right\}\nabla f(x^k, s_k), \tag{SPS$_{\max}$}$$

called the $\text{SPS}_{\max}$ method, where $c, \gamma_b > 0$. The stochastic Polyak step size is closely related to stochastic model-based proximal point methods as well as stochastic bundle methods (Asi & Duchi, 2019; Paren et al., 2022; Schaipp et al., 2023).

*Contributions.* Our proposed method `MoMo` can be seen as an extension of the Polyak step size that also incorporates momentum. This follows from the viewpoint of the Polyak step size (Berrada et al., 2020; Paren et al., 2022; Schaipp et al., 2023) as a truncated model-based method. In particular `MoMo` with no momentum is equal to $\text{SPS}_{\max}$.

**Numerical findings.** We find that `MoMo` consistently improves the sensitivity with respect to hyperparameter choice as compared to `SGD-M` for standard image classification tasks including `MNIST`, `CIFAR10`, `CIFAR100` and `Imagenet`. The same is true for `MoMo-Adam` compared to `Adam` on encoder-decoder transformers on the translation task `IWSLT14`.

Furthermore, we observe that the adaptive learning rate of `MoMo(-Adam)` for some tasks automatically performs a warm-up at the beginning of training and a decay in later iterations, two techniques often used in order to improve training (Sun, 2020).

## 2 MODEL-BASED MOMENTUM METHODS

Let us recall the `SGD` model in (4) which has two issues: First, it approximates a single stochastic function $f(x, s_k)$, as opposed to the full loss $f(x)$. Second, this model can be negative even though our loss function is always positive. Here, we develop a model directly for $f(x)$, and not $f(x, s_k)$, which also takes into account lower bounds on the function value.

### 2.1 MODEL-BASED VIEWPOINT OF MOMENTUM

Suppose we have sampled inputs $s_1, \ldots, s_k$ and past iterates $x^1, \ldots, x^k$. We can use these samples to build a better model of $f(x)$ by averaging past function evaluations as follows

$$f(x) = \mathbb{E}_{s \sim \mathcal{D}}[f(x, s)] \approx \frac{1}{\rho_k}\sum_{j=1}^{k}\rho_{j,k}f(x, s_j), \tag{5}$$

where $\rho_{j,k} \geq 0$ and $\rho_k := \sum_{j=1}^{k}\rho_{j,k}$. Thus, the $\rho_k^{-1}\rho_{j,k}$ are a discrete probability mass function over the previous samples. The issue with (5) is that it is expensive to

evaluate $f(x, s_j)$ for $j = 1, \ldots, k$, which we would need to do at every iteration. Instead, we approximate each $f(x, s_j)$ by linearizing $f(x, s_j)$ around $x^j$, the point it was last evaluated

$$f(x, s_j) \approx f(x^j, s_j) + \langle \nabla f(x^j, s_j), x - x^j \rangle, \quad \text{for } j = 1, \ldots, k. \tag{6}$$

Using (5) and the linear approximations in (6) we can approximate $f(x)$ as follows

$$f(x) \approx \frac{1}{\rho_k} \sum_{j=1}^k \rho_{j,k} \big( f(x^j, s_j) + \langle \nabla f(x^j, s_j), x - x^j \rangle \big) = m_k(x). \tag{7}$$

If we use the above model $m_k(x)$ in (3), then the resulting update is SGD-M

$$x^{k+1} = x^k - \frac{\alpha_k}{\rho_k} d_k, \quad \text{where} \quad d_k := \sum_{j=1}^k \rho_{j,k} \nabla f(x^j, s_j). \tag{8}$$

This gives a new viewpoint of momentum. Next we incorporate a lower bound into this model so that, much like the loss function, it cannot become negative.

## 2.2 Deriving MoMo

Since we know the loss is lower-bounded by zero, we will also impose a lower bound on the model (7). Though we could use zero, we will use an estimate $f_*^k \geq 0$ of the lower bound to allow for cases where $f(x^*)$ may be far from zero. Imposing a lower bound of $f_*^k$ gives the following model

$$f(x) \approx \max \left\{ \frac{1}{\rho_k} \sum_{j=1}^k \rho_{j,k} \big( f(x^j, s_j) + \langle \nabla f(x^j, s_j), x - x^j \rangle \big), f_*^k \right\} =: m_k(x). \tag{9}$$

For overparametrized machine-learning models the minimum value $f(x^*)$ is often close to zero (Ma et al., 2018; Gower et al., 2021). Thus, choosing $f_*^k = 0$ in every iteration will work well (as we verify later in our experiments). For tasks where $f_*^k = 0$ is too loose of a bound, in Section 4 we develop an online estimate for $f_*^k$ based on available information. Using the model (9), we can now define the proximal update

$$x^{k+1} = \underset{y \in \mathbb{R}^d}{\text{argmin}} \; m_k(y) + \frac{1}{2\alpha_k} \|y - x^k\|^2. \tag{10}$$

Because $m_k(y)$ is a simple piece-wise linear function, the update (10) has a closed form solution, as we show in the following lemma (proof in Appendix C.1).

**Lemma 2.1.** [MoMo update] Let

$$d_k := \sum_{j=1}^k \rho_{j,k} \nabla f(x^j, s_j), \quad \bar{f}_k := \sum_{j=1}^k \rho_{j,k} f(x^j, s_j), \quad \gamma_k := \sum_{j=1}^k \rho_{j,k} \langle \nabla f(x^j, s_j), x^j \rangle. \tag{11}$$

Using model (9), the closed form solution to (10) is

$$x^{k+1} = x^k - \tau_k d_k, \quad \tau_k := \min \left\{ \frac{\alpha_k}{\rho_k}, \frac{\left( \bar{f}_k + \langle d_k, x^k \rangle - \gamma_k - \rho_k f_*^k \right)_+}{\|d_k\|^2} \right\}. \tag{12}$$

Finally, it remains to select the averaging coefficients $\rho_{j,k}$. Here we will use an exponentially weighted average that places more weight on recent samples. Aside from working well in practice on countless real-world examples, exponential averaging can be motivated through the model-based interpretation. Recent iterates will most likely have gradients, and loss values, that are closer to our current iterate $x^k$. Thus we place more weight on recent iterates i.e. $\rho_{j,k}$ big for $j$ close to $k$. We give two options for exponentially weighted averaging next.

## 2.3 The Coefficients $\rho_{j,k}$: To bias or not to bias

We now choose $\rho_{j,k} \geq 0$ such that we can update $\bar{f}_k$, $d_k$ and $\gamma_k$ in (11) on the fly, storing only two scalars and one vector, and with the same resulting iteration complexity as SGD-M.

**Exponentially Weighted Average.** Let $\beta \in [0, 1)$. Starting with $\rho_{1,1} = 1$, and for $k \geq 2$ define $\rho_{j,k} = \beta \rho_{j,k-1}$ for $j \leq k - 1$ and $\rho_{j,k} = 1 - \beta$ for $j = k$. Then, $\rho_k = \sum_{j=1}^k \rho_{j,k} = 1$ for

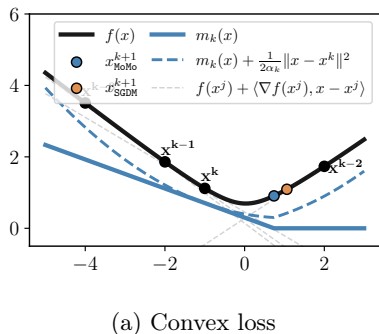

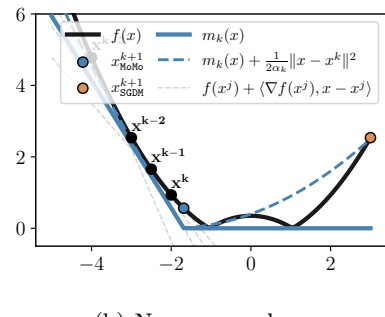

| (a) Convex loss | (b) Non-convex loss |

Figure 1: Illustration of the `MoMo` model (blue curves) for two different loss functions with $\alpha_k = 5$. Due to truncation, the new iterate of `MoMo` (blue point) is closer to the minimum than `SGD-M` (orange point). The right plot shows how `MoMo` takes a small step when gradients are steep, whereas `SGD-M` takes a large step and ends up far from the solution.

all $k \in \mathbb{N}$ and the quantities in (11) are exponentially weighted averages, see Lemma A.1. As a consequence, we can update $\bar{f}_k$, $d_k$ and $\gamma_k$ on the fly as given in lines 4–6 in Algorithm 1. Combining update (12) and the fact that $\rho_k = 1$, we obtain Algorithm 1, which we call `MoMo`.

---

**Algorithm 1:** `MoMo`: Model-based Momentum method.

1 **Default settings:** $\alpha_k = 1$, $\beta = 0.9$, $(f_*^k)_{k \in \mathbb{N}} = 0$.
  **Input:** $x^1 \in \mathbb{R}^d$, $\beta \in [0, 1)$, $\alpha_k > 0$, $(f_*^k)_{k \in \mathbb{N}} \subset \mathbb{R}$
2 **Init:** $\bar{f}_0 = f(x^1, s_1), d_0 = \nabla f(x^1, s_1), \gamma_0 = \langle d_0, x^1 \rangle$
3 **for** $k = 1$ **to** $K - 1$ **do**
4   $\bar{f}_k = (1 - \beta)f(x^k, s_k) + \beta\bar{f}_{k-1}$
5   $\gamma_k = (1 - \beta)\langle \nabla f(x^k, s_k), x^k \rangle + \beta\gamma_{k-1}$
6   $d_k = (1 - \beta)\nabla f(x^k, s_k) + \beta d_{k-1}$
7   $h_k = \bar{f}_k + \langle d_k, x^k \rangle - \gamma_k$
8   $x^{k+1} = x^k - \min\left\{\alpha_k, \frac{(h_k - f_*^k)_+}{\|d_k\|^2}\right\}d_k$
  **Output:** $x^K$

**Remark 2.2.** The *adaptive learning rate* $\tau_k$ in (12) determines the size of the step and can vary in each iteration even if $\alpha_k$ is constant. The *(user-specified) learning rate* $\alpha_k$ caps the adaptive learning rate.

**Remark 2.3** (Complexity). `MoMo` has the same order iteration complexity and memory footprint as `SGD-M`. `MoMo` stores two additional scalars $\gamma_k$ and $\bar{f}_k$, as compared to `SGD-M`, and has two additional $\mathcal{O}(d)$ inner products lines 5 and 7, and one $\mathcal{O}(d)$ vector norm on line 8.

---

For $\beta = 0$ (no momentum), we have $\gamma_k = \langle \nabla f(x^k, s_k), x^k \rangle = \langle d_k, x^k \rangle$ and $\bar{f}_k = f(x^k, s_k)$. Consequently $h_k = f(x^k, s_k)$, and in this special case, `MoMo` is equivalent[3] to (SPS$_{\max}$).

Fig. 1 shows how the `MoMo` model (10) approximates a convex function (left) and a non-convex function (right). The `MoMo` update $x_{\text{MoMo}}^{k+1}$ in Fig. 1 is closer to the minima (left) and sometimes much closer (right) on non-convex problems, as compared the `SGD-M` update.

**Averaging with Bias Correction.** Alternatively, we can choose $\rho_{j,k} = (1 - \beta)\beta^{k-j}$ for $j = 1, \ldots, k$, as it is used in `Adam` (Kingma & Ba, 2015). This gives $\rho_k = 1 - \beta^k \neq 1$. We discuss this choice for `MoMo` in Appendix A.1 and will use it later for `MoMo-Adam`.

## 3   WEIGHT DECAY AND PRECONDITIONING

Often weight decay is used in order to improve generalization (Zhang et al., 2019). Weight decay is equivalent to adding a squared $\ell_2$-regularization to the objective function (Krogh & Hertz, 1991), in other words, instead of (1) we solve $\min_{x \in \mathbb{R}^d} f(x) + \frac{\lambda}{2}\|x\|^2$, where $f(x)$ is again the loss function. To include weight decay, we build a model $m_k$ for the loss $f$ and keep the $\ell_2$-regularization outside of the model. That is equation (10) is modified to

$$x^{k+1} = \operatorname*{argmin}_{y \in \mathbb{R}^d} m_k(y) + \frac{\lambda}{2}\|y\|^2 + \frac{1}{2\alpha_k}\|y - x^k\|^2. \tag{13}$$

---

[3]This equivalence requires setting $\gamma_b \leftarrow \alpha_k$, $c \leftarrow 1$, and assuming $f_*^k = \inf_z f(z, s_k)$.

**Algorithm 2:** `MoMo-Adam`: Adaptive learning rates for `Adam`
___
1 **Default settings:** $\alpha_k = 10^{-2}$, $(\beta_1, \beta_2) = (0.9, 0.999)$, $\epsilon = 10^{-8}$
   **Input:** $x^1 \in \mathbb{R}^d$, $\beta_1, \beta_2 \in [0, 1)$, $\epsilon > 0$, $\alpha_k > 0$, $\lambda \geq 0$, and $(f_*^k)_{k \in \mathbb{N}} \subset \mathbb{R}$.
2 **Initialize:** $\bar{f}_0 = 0, d_0 = 0, \gamma_0 = 0$, and $v_0 = 0$.
3 **for** $k = 1$ **to** $K - 1$ **do**
4    $g_k = \nabla f(x^k, s_k)$;   $d_k = (1 - \beta_1)g_k + \beta_1 d_{k-1}$
5    $v_k = \beta_2 v_{k-1} + (1 - \beta_2)(g_k \odot g_k)$
6    $\mathbf{D}_k = \text{Diag}\big(\epsilon \mathbf{1}_d + \sqrt{v_k/(1 - \beta_2^k)}\big)$
7    $\bar{f}_k = (1 - \beta_1)f(x^k, s_k) + \beta_1 \bar{f}_{k-1}$
8    $\gamma_k = (1 - \beta_1)\langle g_k, x^k \rangle + \beta_1 \gamma_{k-1}$
9    $\tau_k = \min\left\{ (1-\beta_1^k)^{-1}\alpha_k, \big((1+\lambda\alpha_k)(\bar{f}_k - \gamma_k - (1-\beta_1^k)f_*^k) + \langle d_k, x^k \rangle\big)_+ / \|d_k\|_{\mathbf{D}_k^{-1}}^2 \right\}$
10    $x^{k+1} = \frac{1}{1+\alpha_k\lambda}\big[ x^k - \tau_k \mathbf{D}_k^{-1} d_k \big]$
   **Output:** $x^K$
___

Finally, the Euclidean norm may often not be best suited. Many popular methods such as `AdaGrad` or `Adam` are based on using a preconditioner for the proximal step. Hence, we allow for an arbitrary norm defined by a symmetric, positive definite matrix $\mathbf{D}_k \in \mathbb{R}^{d \times d}$, i.e. $\|x\|_{\mathbf{D}_k}^2 := \langle \mathbf{D}_k x, x \rangle$. We can now use $\mathbf{D}_k$ to change the metric within our proximal method

$$x^{k+1} = \underset{y \in \mathbb{R}^d}{\arg\min}\, m_k(y) + \frac{\lambda}{2}\|y\|_{\mathbf{D}_k}^2 + \frac{1}{2\alpha_k}\|y - x^k\|_{\mathbf{D}_k}^2. \tag{14}$$

This update (14) enjoys the following closed form solution (proof in Appendix C.2).

**Lemma 3.1.** Using model (9), the closed form solution to (14) is given by

$$\tau_k = \min\left\{ \frac{\alpha_k}{\rho_k}, \frac{\big((1+\alpha_k\lambda)(\bar{f}_k - \rho_k f_*^k - \gamma_k) + \langle d_k, x^k \rangle\big)_+}{\|d_k\|_{\mathbf{D}_k^{-1}}^2} \right\}, \tag{15}$$

$$x^{k+1} = \frac{1}{1+\alpha_k\lambda}\big[ x^k - \tau_k \mathbf{D}_k^{-1} d_k \big]. \tag{16}$$

Lemma 3.1 shows how to incorporate weight decay in `MoMo`: we replace Line 8 in Algorithm 1 by (16) with $\mathbf{D}_k = \mathbf{Id}$ and $\rho_k = 1$. If $\beta = 0$ (no momentum) then `MoMo` with weight decay recovers `ProxSPS`, the proximal version of the stochastic Polyak step (Schaipp et al., 2023).

**Deriving `MoMo-Adam`.** Using Lemma 3.1 we can obtain an `Adam`-version of `MoMo` by defining $\mathbf{D}_k$ as the diagonal preconditioner of `Adam`. Let $\mathbf{1}_d$ be the $d$-dimensional vector of ones, $\text{Diag}(v)$ a diagonal matrix with diagonal entries $v \in \mathbb{R}^d$, and $\odot$ and $\sqrt{v}$ the elementwise multiplication and square-root operations. Denoting $g_k = \nabla f(x^k, s_k)$, we choose

$$v_k = (1 - \beta_2)v_{k-1} + \beta_2(g_k \odot g_k), \quad \mathbf{D}_k = \text{Diag}(\epsilon \mathbf{1}_d + \sqrt{v_k/(1 - \beta_2)^k}),$$

where $\beta_2 \in [0, 1)$, $\epsilon > 0$. Using this preconditioner with Lemma 3.1 gives Algorithm 2, called `MoMo-Adam`. Note that here we choose $\rho_{j,k} = (1 - \beta)\beta^{k-j}$ (cf. Section 2.3) which gives the standard averaging scheme of `Adam`. We focus on `MoMo` versions of `SGD-M` and `Adam` because these are the two most widely used methods. However, from Lemma 3.1 we could easily obtain a `MoMo`-version of different variations, such as `Adabelief` (Zhuang et al., 2020).

## 4   Estimating a Lower Bound

So far, we have assumed that lower-bound estimates $(f_*^k)$ are given with $f_*^k = 0$ being the default. However, this might not be a tight estimate of $f^*$ (e.g. when training transformers). In such situations, we derive an online estimate of the lower bound. In particular, for convex functions we will derive a lower bound for an unbiased estimate of $f(x^*)$ given by

$$\bar{f}_*^k := \frac{1}{\rho_k}\sum_{j=1}^k \rho_{j,k}f(x^*, s_j), \quad \text{where} \quad \mathbb{E}\big[\bar{f}_*^k\big] = f(x^*). \tag{17}$$

Though $\bar{f}_*^k$ is not equal to $f(x^*)$, it is an unbiased estimate since $\mathbb{E}\left[f(x^*, s_j)\right] = f(x^*)$. It is also a reasonable choice since we motivated our method using the analogous approximation of $f(x)$ in (5). Furthermore, if $f_*^k = \bar{f}_*^k$ then for any preconditioner and convex losses, an iterate of MoMo can only decrease the distance to a given optimal point, as we show next.

**Lemma 4.1.** Let $f(\cdot, s)$ be convex for every $s$ and let $x^* \in \arg\min_{x \in \mathbb{R}^d} f(x)$. For the iterates of the general MoMo update (cf. Lemma 3.1) with $\lambda = 0$ and $f_*^k = \bar{f}_*^k$, it holds

$$\left\|x^{k+1} - x^*\right\|_{\mathbf{D}_k}^2 \le \left\|x^k - x^*\right\|_{\mathbf{D}_k}^2 - \tau_k(h_k - \rho_k \bar{f}_*^k)_+. \tag{18}$$

We use this monotonicity to derive a convergence theorem for MoMo in Theorem F.2. The following lemma derives an estimate $f_*^k \ge 0$ for $\bar{f}_*^k$ given in (17) by using readily available information for any momentum-based method, such as Algorithm 2.

**Lemma 4.2.** Let $f(x, s)$ be convex in $x$ for all $s \in \mathcal{D}$. Let $x^k$ be given by (16) with $\lambda = 0$. Let $\eta_k := \prod_{j=2}^k \lambda_{\min}\left(\mathbf{D}_j^{-1}\mathbf{D}_{j-1}\right)$, and $h_k := \bar{f}_k + \langle d_k, x^k \rangle - \gamma_k$. We have $\bar{f}_*^k \ge f_*^{k+1}$ where

$$f_*^{k+1} := \frac{1}{2\eta_k \tau_k \rho_k}\left(\sum_{j=1}^k 2\eta_j \tau_j \left(h_j - \frac{1}{2}\tau_j \|d_j\|_{\mathbf{D}_j^{-1}}^2\right) - D_1^2 - 2\sum_{j=1}^{k-1} \eta_j \tau_j \rho_j \bar{f}_*^j\right)$$

where $D_1 := \left\|x^1 - x^*\right\|_{\mathbf{D}_1}$. Bootstrapping by using $f_*^k \approx \bar{f}_*^{k-1}$ we have for $k \ge 2$ that

$$f_*^{k+1} = \frac{1}{\rho_k}\left(h_k - \frac{1}{2}\tau_k \|d_k\|_{\mathbf{D}_k^{-1}}^2\right). \tag{19}$$

To simplify the discussion, consider the case without a preconditioner, i.e. $\mathbf{D}_k = \mathbf{Id}$, thus $\eta_k = 1$. First, note that $f_*^{k+1}$ depends on the initial distance to the solution $D_1$, which we do not know. Fortunately, $D_1$ does not appear in the recursive update (19), because it only appears in $f_*^1$. We can circumvent this initial dependency by simply setting $f_*^1 = 0$.

We need one more precautionary measure, because we cannot allow the step size $\tau_k$ in (15) to be zero. That is, by examining (15) we have to disallow that

$$(1 + \alpha_k \lambda)\rho_k f_*^k \ge (1 + \alpha_k \lambda)(f_*^k - \gamma_k) + \langle d_k, x^k \rangle =: h_k^\lambda. \tag{20}$$

Hence, in each iteration of MoMo or MoMo-Adam, we call the ResetStar routine in Algorithm 3 *before* the update of $x^{k+1}$ that checks if this upper bound has been crossed, and if so, resets $f_*^k$ to be sufficiently small. After updating $x^{k+1}$, we update $f_*^{k+1}$ with EstimateStar routine in Algorithm 4, according to Lemma 4.2. We call the respective methods MoMo$^*$ and MoMo-Adam$^*$. For completeness, we give the full algorithm of MoMo$^*$ in Algorithm 6 in the Appendix. We give an example of how the values of $f_*^k$ converge to $f^*$ in Appendix E.4.

| **Algorithm 3: ResetStar** | **Algorithm 4: EstimateStar** |
|---|---|
| **Input:** $f_*^k, \alpha_k, \lambda, \rho_k, h_k^\lambda$ | **Input:** $\bar{f}_k, x^k, \gamma_k, \tau_k, d_k, \mathbf{D}_k, \rho_k$ |
| 1 **if** (20) **then** | 1 $h_k = \bar{f}_k + \langle d_k, x_k \rangle - \gamma_k$ |
| 2 $\quad \lfloor\ f_*^k = \max\left\{\frac{1}{2}[(1 + \alpha_k \lambda)\rho_k]^{-1}h_k^\lambda, f_*^1\right\}$ | 2 $f_*^{k+1} = \max\left\{\rho_k^{-1}(h_k - \frac{1}{2}\tau_k \|d_k\|_{\mathbf{D}_k^{-1}}^2, f_*^1\right\}$ |
| **Output:** $f_*^k$ | **Output:** $f_*^{k+1}$ |

## 5 EXPERIMENTS

Our experiments will focus on the sensitivity with respect to choice of the learning rate $\alpha_k$. Schmidt et al. (2021) showed that most optimization methods perform equally well when being tuned. For practical use a tuning budget needs to be considered, and hence we are interested in methods that require little or no tuning. Here we investigate how using our MoMo adaptive learning rate can improve the stability of both SGD-M and Adam. To do this, for each task and model, we do a learning-rate sweep for both SGD-M, Adam, MoMo and MoMo-Adam and compare the resulting validation score for each learning rate.

For MoMo and MoMo-Adam, note that the effective step size (cf. (16)) has the form

$$\tau_k = \min\{\tfrac{\alpha_k}{\rho_k}, \zeta_k\} \quad \text{with} \quad \zeta_k := \tfrac{1}{\|d_k\|_{\mathbf{D}_k^{-1}}^2}\left((1 + \alpha_k \lambda)(\bar{f}_k - \rho_k f_*^k - \gamma_k) + \langle d_k, x^k \rangle\right)_+. \tag{21}$$

We refer to Algorithm 1, Line 8 and Algorithm 2, Line 10 for the exact formula for MoMo and MoMo-Adam (For MoMo we have that $\rho_k = 1, \mathbf{D}_k = \mathbf{Id}$). We will refer to $\alpha_k$ as the *(user-specified) learning rate* and to $\tau_k$ as the *adaptive learning rate*.

## 5.1 ZERO AS LOWER BOUND

First, we compare the `MoMo` methods to `SGD-M` and `Adam` for problems where zero is a good estimate of the optimal value $f^*$. In this section, we set $f_*^k = 0$ for all $k \in \mathbb{N}$ for `MoMo(-Adam)`.

**Models and Datasets.** We do the following tasks (more details in Appendix E.3).

- `ResNet110` for `CIFAR100`, `ResNet20`, `VGG16`, and `ViT` for `CIFAR10`
- `DLRM` for `Criteo` Kaggle Display Advertising Challenge,
- `MLP` for `MNIST`: two hidden layers of size 100 and `ReLU`.

**Parameter Settings.** We use default choices for momentum parameter $\beta = 0.9$ for `MoMo` and `SGD-M`, and $(\beta_1, \beta_2) = (0.9, 0.999)$ for `MoMo-Adam` and `Adam` respectively. In the experiments of this section, we always report averaged values over three seeds (five for `DLRM`).

**Discussion.** We run `MoMo`, `MoMo-Adam`, `Adam` and `SGD-M`, for a fixed number of epochs (cf. Appendix E.3), using a constant learning rate $\alpha_k = \alpha_0$. The plots in Fig. 2 show the final training loss (top) and accuracy on the validation set (bottom) of each method when varying the learning rate $\alpha_0$. The training curves for the best runs can be found in Figs. E.1 and E.2. For `VGG16` for `CIFAR10` and `MLP` for `MNIST`, the same plots can be found in Appendix E. We observe that for small learning rates `MoMo` (`MoMo-Adam`) is identical to `SGD-M` (`Adam`). This is expected, since for small $\alpha_0$, we have $\tau_k = \alpha_0$ (see (21)).

For larger learning rates, we observe that `MoMo` and `MoMo-Adam` improve the training loss and validation accuracy, but `SGD-M` and `Adam` decline in performance or even fail to converge. Most importantly, `MoMo(-Adam)` consistently extends the range of "good" learning rates by over one order of magnitude. Further, `MoMo(-Adam)` achieve the overall best validation accuracy for all problems except `DLRM` and `ViT`, where the gap to the best score is minute and within the standard deviation of running multiple seeds.

This advantage can be explained with the adaptivity of the step size of `MoMo(-Adam)`. In Fig. E.4a, we plot the adaptive term $\zeta_k$ (21) for `MoMo` on a `ResNet20`. For $\alpha_0 \in [1, 10]$, we observe that the effective learning rate $\tau_k$ is adaptive even though $\alpha_k$ is constant. We observe two phenomena: firstly, in Fig. E.4a `MoMo` is doing an automatic learning rate decay *without any user choice for a learning-rate schedule*. Secondly, in the very first iterations, `MoMo` is doing a warm-up of the learning rate as $\tau_k = \zeta_k$ starts very small, but quickly becomes large. Both dynamics of $\tau_k$ help to improve performance and stability. We also observe faster initial training progress of `MoMo(-Adam)` (cf. Figs. E.1 and E.2).

For all of the above tasks, the (training) loss converges to values below 0.5. Next, we consider two problems where the final training loss is significantly above zero. In such situations, we find that `MoMo` methods with $f_*^k = 0$ are less likely to make use of the adaptive term $\zeta_k$. As a consequence, `MoMo` with $f_*^k = 0$ will yield little or no improvement. To see improvement, we employ the online estimation of a lower bound for `MoMo` given in Lemma 4.2.

## 5.2 ONLINE LOWER BOUND ESTIMATION

We now consider image classification on `Imagenet32/-1k` and a transformer for German-to-English translation. For both problems, the optimal value $f^*$ is far away from zero and hence we use `MoMo` with a known estimate of $f^*$ or with the online estimation developed in Section 4. Details on models and datasets are listed in Appendix E.3.

**Imagenet Classification.** We train a `ResNet18` for `Imagenet32` and give the resulting validation accuracy in Fig. 3a for weight decay $\lambda = 0$. We show the results $\lambda = 10^{-4}$ and for `Imagenet-1k` in the appendix in Fig. E.5. We run `MoMo(-Adam)` first with constant lower bound $f_*^k = 0$ and an *oracle* value $f_*^k = 0.9$. Further, we run `MoMo(-Adam)*` (indicated by the suffix *-star* in the plots), (cf. Algorithm 6). We compare to `SGD-M` and `AdamW` as baseline. For all methods, we use a constant learning rate $\alpha_k = \alpha_0$ and vary the value of $\alpha_0$.

First, observe that lower bound $f_*^k = 0$ leads to similar performance as the baseline method (in particular it is never worse). Next, observe that the tighter lower bound $f_*^k = 0.9$ leads to improvement for all learning rates. Finally, the online estimated lower bound widens the range of learning rate with good accuracy by an order of magnitude and leads to small improvements in top accuracy.

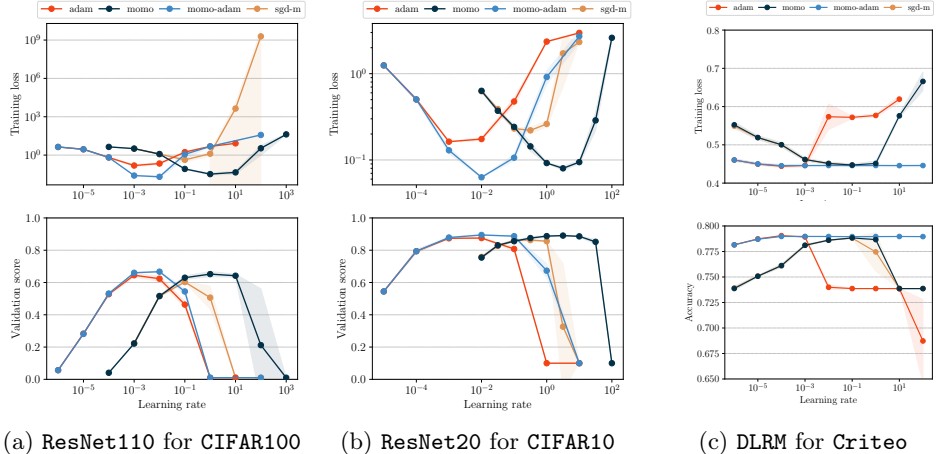

Figure 2: Training loss (top row) and validation accuracy (bottom row) after a fixed number of epochs, for varying (constant) learning rate $\alpha_0$. Shaded area depicts two standard deviations.

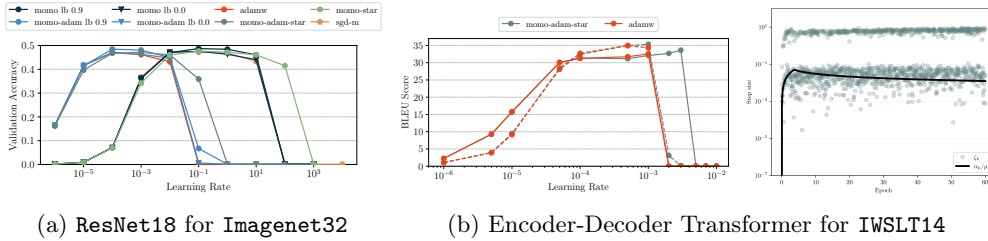

Figure 3: Validation accuracy over a range of learning rates $\alpha_0$. (a) `Imagenet32` without weight decay ($\lambda = 0$). (b) Left: `IWSLT14` translation task with dropout 0.1 (**plain**) or 0.3 (**dashed**). Right: Learning rate schedule (**black**) and adaptive step sizes (**grey dots**) of `MoMo-Adam`* for $\alpha_0 = 5 \cdot 10^{-2}$.

**Transformer for German-to-English Translation.** We consider the task of neural machine translation from German to English by training an encoder-decoder transformer architecture (Vaswani et al., 2017) on the `IWSLT14` dataset. We run two settings, namely dropout of 0.1 and 0.3. We fine-tune the hyperparameters of the baseline `AdamW`: for the learning-rate schedule $\alpha_k$, we use a linear warm-up of 4000 iterations from zero to a given value $\alpha_0$ followed by an inverse square-root decay (cf. Fig. 3b for an example curve and the adaptive step sizes). All other parameter settings are given in Appendix E.3. `MoMo-Adam`* uses the same hyperparameter settings as `AdamW`.

Fig. 3b shows the BLEU score after 60 epochs when varying the initial learning rate $\alpha_0$: `MoMo-Adam`* is on par or better than `AdamW` on the full range of initial learning rates and for both dropout values. While the improvement is not as substantial as for previous examples, we remark that for this particular task we compare to a fine-tuned configuration of `AdamW`.

## 6 Conclusion

We present `MoMo` and `MoMo-Adam`, adaptive learning rates for `SGD-M` and `Adam`. The main conceptual insight is that momentum can be used to build a model of the loss by averaging a stream of loss function values and gradients. Combined with truncating this average at a known lower bound of the loss, we obtain the `MoMo` algorithms. This technique can be applied potentially to other methods, for example variants of `Adam`.

We show examples where incorporating `MoMo` into `SGD-M` and `Adam` significantly reduces the sensitivity to learning rate choice. This can be particularly helpful for practitioners who look for good out-of-the-box optimization performance for new tasks.

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

CONTENTS

## A  Implementation details

### A.1  Notes on the Averaging Coefficients

**Lemma A.1.** Let $\beta \in [0, 1)$. Let $\rho_{1,1} = 1$, and for $k \geq 2$ let

$$\rho_{j,k} = \begin{cases} \beta \rho_{j,k-1}, & j \leq k - 1, \\ 1 - \beta, & j = k. \end{cases}$$

Then, $\sum_{j=1}^{k} \rho_{j,k} = 1$ holds for all $k \in \mathbb{N}$. Further, for an arbitrary sequence $(u_j)_{j \in \mathbb{N}} \subset \mathbb{R}^m$, $m \in \mathbb{N}$, consider the weighted sum

$$\bar{u}_k := \sum_{j=1}^{k} \rho_{j,k} u_j.$$

Then, if $\bar{u}_0 := u_1$ it holds $\bar{u}_k = (1 - \beta) u_k + \beta \bar{u}_{k-1}$ for all $k \in \mathbb{N}$.

*Proof.* We prove that $\sum_{j=1}^{k} \rho_{j,k} = 1$ holds for all $k \in \mathbb{N}$ by induction. For the base case $k = 1$, we have $\rho_{1,1} = 1$ by definition. Assuming that $\sum_{j=1}^{k-1} \rho_{j,k-1} = 1$, we have

$$\sum_{j=1}^{k} \rho_{j,k} = \rho_{k,k} + \sum_{j=1}^{k-1} \rho_{j,k} = 1 - \beta + \beta \sum_{j=1}^{k-1} \rho_{j,k-1} = 1 - \beta + \beta = 1.$$

Consequently, we have $\bar{u}_1 = \rho_{11} u_1 = u_1$, and for $k \geq 2$,

$$\bar{u}_k = \sum_{j=1}^{k} \rho_{j,k} u_j = (1 - \beta) u_k + \sum_{j=1}^{k-1} \beta \rho_{j,k-1} u_j = (1 - \beta) u_k + \beta \sum_{j=1}^{k-1} \rho_{j,k-1} u_j$$
$$= (1 - \beta) u_k + \beta \bar{u}_{k-1}.$$

$\square$

For the choice of $\rho_{j,k}$ in Lemma A.1, unrolling the recursion, for $k \geq 2$ we obtain the explicit formula

$$\rho_{j,k} = \begin{cases} (1 - \beta) \beta^{k-j}, & j \geq 2 \\ \beta^{k-1}, & j = 1. \end{cases} \tag{22}$$

**Averaging with Bias Correction.**  Chosing $\rho_{j,k} = (1 - \beta) \beta^{k-j}$, we have $\rho_{j,k} = \beta \rho_{j,k-1}$, and $\rho_{k,k} = 1 - \beta$. Hence, we can update $\bar{f}_k = (1 - \beta) f(x^k, s_k) + \beta \bar{f}_{k-1}$ and analogously for $d_k, \gamma_k$. However, this choice does not satisfy $\sum_{j=1}^{k} \rho_{j,k} = 1$. Indeed using the geometric series gives

$$\rho_k = (1 - \beta) \sum_{j=0}^{k-1} \beta^j = 1 - \beta^k.$$

This fact motivates scaling by the factor of $1 - \beta^k$ which was termed *debiasing* in `Adam`. This alternative averaging scheme leads to a variant of `MoMo` with bias correction, presented in Algorithm 5. As the two presented choices of $\rho_{j,k}$ are very similar, we do not expect major differences in their performance (cf. Remark A.2).

**Remark A.2.** Algorithm 5 differs from Algorithm 1 only in two steps: first, the quantities $\bar{f}_0$, $d_0$, $\gamma_0$ are initialized at zero. Secondly, we use $\frac{\alpha_k}{1 - \beta^k}$ instead of $\alpha_k$ and $(1 - \beta^k) f_*^k$ instead of $f_*^k$ in line (6). As $\beta \in [0, 1)$, for late iteration number $k$, we can expect that both methods behave very similarly.

---

**Algorithm 5:** `MoMo-Bias`: Model-based Momentum with bias correction. Defaults settings $\beta = 0.9$.

---

**Input:** $x^1 \in \mathbb{R}^d$, $\beta \in [0, 1)$, $\alpha_k > 0$, $(f_*^k)_{k \in \mathbb{N}} \subset \mathbb{R}$.

**1 Initialize:** $\bar{f}_0 = 0, d_0 = 0$ and $\gamma_0 = 0$.

**2 for** $k = 1$ **to** $K - 1$ **do**

**3** $\quad \bar{f}_k = (1 - \beta)f(x^k, s_k) + \beta \bar{f}_{k-1}$

**4** $\quad d_k = (1 - \beta)\nabla f(x^k, s_k) + \beta d_{k-1}$

**5** $\quad \gamma_k = (1 - \beta)\langle \nabla f(x^k, s_k), x^k \rangle + \beta \gamma_{k-1}$

**6** $\quad x^{k+1} = x^k - \min\left\{ \dfrac{\alpha_k}{1 - \beta^k}, \dfrac{(\bar{f}_k - (1 - \beta^k)f_*^k + \langle d_k, x^k \rangle - \gamma_k)_+}{\|d_k\|^2} \right\} d_k.$

**Output:** $x^K$

---

### A.2 COMPARISON OF `MoMo-Adam` TO `AdamW`

Algorithm 2 naturally compares to `AdamW` (Loshchilov & Hutter, 2019). Note that the update of `AdamW` (in the notation of Algorithm 2) can be written as

$$x^{k+1} = (1 - \alpha_k \lambda)x^k - \frac{\alpha_k}{1 - \beta_1^k}\mathbf{D}_k^{-1}d_k,$$

Compared to Algorithm 2, Line 10, the weight decay of `AdamW` is not done dividing the whole expression by $\frac{1}{1 + \alpha_k \lambda}$, but instead multiplying only $x^k$ with $1 - \alpha_k \lambda$. This is a first-order Taylor approximation (Zhuang et al., 2022): for $\alpha$ small it holds $\frac{1}{1 + \alpha \lambda} \approx 1 - \alpha \lambda$ and $\frac{\alpha}{1 + \alpha \lambda} \approx \alpha$. If we would want to adapt this approximation, we could replace Line 10 with

$$x^{k+1} = (1 - \lambda \alpha_k)x^k - \min\left\{ \frac{\alpha_k}{1 - \beta_1^k}, \frac{\left((1 + \lambda \alpha_k)(\bar{f}_k - (1 - \beta_1^k)f_*^k - \gamma_k) + \langle d_k, x^k \rangle\right)_+}{\|d_k\|_{\mathbf{D}_k^{-1}}^2} \right\}\mathbf{D}_k^{-1}d_k. \tag{23}$$

However, the results of (Zhuang et al., 2022) suggest that this approximation has almost no impact on the empirical performance.

### A.3 `MoMo`*

Here we give the complete pseudocode for `MoMo`*, that is the `MoMo` method that uses the estimator for $f_*^k$ given in Lemma 4.2.

---

**Algorithm 6:** `MoMo`*: Adaptive learning rates and online estimation of $f^*$.

---

**Input:** $x^1 \in \mathbb{R}^d$, $\beta \in [0, 1)$, $\alpha_k > 0$, $f_*^1 \subset \mathbb{R}$.

**1 Initialize:** $\bar{f}_0 = f(x^1, s_1), d_0 = \nabla f(x^1, s_1)$ and $\gamma_0 = \langle d_0, x^1 \rangle$

**2 for** $k = 1$ **to** $K - 1$ **do**

**3** $\quad \bar{f}_k = (1 - \beta)f(x^k, s_k) + \beta \bar{f}_{k-1}$

**4** $\quad \gamma_k = (1 - \beta)\langle \nabla f(x^k, s_k), x^k \rangle + \beta \gamma_{k-1}$

**5** $\quad d_k = (1 - \beta)\nabla f(x^k, s_k) + \beta d_{k-1}$

**6** $\quad f_*^k = \texttt{ResetStar}()$

**7** $\quad x^{k+1} = x^k - \min\left\{ \alpha_k, \dfrac{(\bar{f}_k - f_*^k + \langle d_k, x^k \rangle - \gamma_k)_+}{\|d_k\|^2} \right\} d_k$

**8** $\quad f_*^{k+1} = \texttt{EstimateStar}().$

**Output:** $x^K$

---

## B AUXILIARY LEMMAS

**Lemma B.1.** Let $y_0, a \in \mathbb{R}^p$ with $a \neq 0$ and $c \in \mathbb{R}$. Let $\beta > 0$. The solution to

$$y^+ = \arg\min_y \underbrace{\left(c + \langle a, y - y_0 \rangle\right)_+}_{:=h(y)} + \frac{1}{2\beta}\|y - y_0\|^2 \tag{24}$$

is given by

$$y^+ = y_0 - \underbrace{\min\left\{\beta, \frac{(c)_+}{\|a\|^2}\right\}}_{:=\tau} a.$$

Moreover we have $h(y^+) = \left(c - \tau\|a\|^2\right)_+$ and

$$h(y^+) = c - \tau\|a\|^2, \quad \text{if } c \geq 0. \tag{25}$$

*Proof.* Clearly, the objective of (24) is strongly convex and therefore there exists a unique solution. The (necessary and sufficient) first-order optimality condition is given by

$$0 = ta + \beta^{-1}(y^+ - y_0), \quad t \in \partial(\cdot)_+(c + \langle a, y^+ - y_0 \rangle). \tag{26}$$

We distinguish three cases:

(P1) Suppose $c < 0$. Then, $y_0$ satisfies (26) with $t = 0$ and hence $y^+ = y_0$. In this case $\tau = 0$ and $h(y^+) = 0 = (c)_+$.

(P2) Let $\bar{y} := y_0 - \beta a$ and assume $c + \langle a, \bar{y} - y_0 \rangle > 0 \iff c - \beta\|a\|^2 > 0 \iff \frac{c}{\|a\|^2} > \beta$. Then $\bar{y}$ satisfies (26) with $t = 1$ and hence $y^+ = \bar{y}$. As $\beta > 0$, hence $c > 0$ and $\tau = \beta$. As $h(y^+) = c + \langle a, y^+ - y_0 \rangle = c - \beta\|a\|^2$, equation (25) holds.

(P3) If neither $c < 0$ nor $\frac{c}{\|a\|^2} > \beta$ hold, then it must hold $c + \langle a, y^+ - y_0 \rangle = 0$. Then, the optimality condition is $0 = ta + \beta^{-1}(y^+ - y_0)$ for some $t \in [0, 1]$. Hence, $y^+ = y_0 - t\beta a$ and $c + \langle a, y^+ - y_0 \rangle = c - t\beta\|a\|^2 = 0 \iff t = \frac{c}{\beta\|a\|^2}$. As $c \geq 0$ we have $t \geq 0$ and $\frac{c}{\|a\|^2} \leq \beta$ implies $t \leq 1$. Hence, $\tau = \frac{c}{\|a\|^2}$ and $c - \tau\|a\|^2 = c - c = 0$, so (25) holds.

$\square$

**Lemma B.2.** Let $y_0, a \in \mathbb{R}^p$ with $a \neq 0$ and $c \in \mathbb{R}$. Let $\mathbf{D} \in \mathbb{R}^{p \times p}$ be a symmetric, positive definite matrix. The solution to

$$y^+ = \underset{y \in \mathbb{R}^p}{\text{argmin}} \underbrace{\left(c + \langle a, y - y_0 \rangle\right)_+}_{:=h(y)} + \frac{1}{2\alpha}\|y - y_0\|_{\mathbf{D}}^2 + \frac{\lambda}{2}\|y\|_{\mathbf{D}}^2 \tag{27}$$

is given by

$$y^+ = \frac{1}{1 + \lambda\alpha}\left[y_0 - \underbrace{\min\left\{\alpha, \frac{\left((1 + \lambda\alpha)c - \lambda\alpha\langle a, y_0 \rangle\right)_+}{\|a\|_{\mathbf{D}^{-1}}^2}\right\}}_{=:\tau} \mathbf{D}^{-1}a\right].$$

Furthermore

$$h(y^+) = \left(c - \frac{\lambda\alpha}{1 + \lambda\alpha}\langle a, y_0 \rangle - \frac{\tau}{1 + \lambda\alpha}\|a\|_{\mathbf{D}^{-1}}^2\right)_+.$$

*Proof.* First we complete the squares as follows

$$\frac{\lambda}{2}\|y\|_{\mathbf{D}}^2 + \frac{1}{2\alpha}\|y - y_0\|_{\mathbf{D}}^2 = \frac{1}{2\alpha}\|y\|_{(1+\lambda\alpha)\mathbf{D}}^2 - \frac{1}{\alpha}\langle y, \mathbf{D}y_0 \rangle + \text{cst.}(y)$$

$$= \frac{1}{2\alpha}\|y\|_{(1+\lambda\alpha)\mathbf{D}}^2 - \frac{1}{\alpha}\langle y, (1+\lambda\alpha)\mathbf{D}\frac{y_0}{1+\lambda\alpha}\rangle + \text{cst.}(y)$$

$$= \frac{1}{2\alpha}\|y - \frac{1}{1+\lambda\alpha}y_0\|_{(1+\lambda\alpha)\mathbf{D}}^2 + \text{cst.}(y),$$

where cst.$(y)$ denotes terms that are constant in $y$. Using the above, (27) is equivalent to

$$y^+ = \operatorname*{argmin}_{y \in \mathbb{R}^p} h(y) + \frac{1}{2\alpha}\|y - \tfrac{1}{1+\lambda\alpha}y_0\|^2_{(1+\lambda\alpha)\mathbf{D}}$$

$$= \operatorname*{argmin}_{y \in \mathbb{R}^p} \left(c + \langle a, y - \tfrac{1}{1+\lambda\alpha}y_0\rangle + \left(\tfrac{1}{1+\lambda\alpha} - 1\right)\langle a, y_0\rangle\right)_+ + \frac{1}{2\alpha}\|y - \tfrac{1}{1+\lambda\alpha}y_0\|^2_{(1+\lambda\alpha)\mathbf{D}}.$$

Let $\hat{c} := c + \left(\tfrac{1}{1+\lambda\alpha} - 1\right)\langle a, y_0\rangle = c - \tfrac{\lambda\alpha}{1+\lambda\alpha}\langle a, y_0\rangle$. With this definition, problem (27) is equivalent to

$$y^+ = \operatorname*{argmin}_{y \in \mathbb{R}^p} \left(\hat{c} + \langle a, y - \tfrac{1}{1+\lambda\alpha}y_0\rangle\right)_+ + \frac{1}{2\alpha}\|y - \tfrac{1}{1+\lambda\alpha}y_0\|^2_{(1+\lambda\alpha)\mathbf{D}}.$$

Changing variables with $z^+ = \mathbf{D}^{1/2}y^+$, $z = \mathbf{D}^{1/2}y$, and $z_0 = \mathbf{D}^{1/2}y_0$ gives

$$z^+ = \operatorname*{argmin}_{z \in \mathbb{R}^p} \left(\hat{c} + \langle \mathbf{D}^{-1/2}a, z - \tfrac{1}{1+\lambda\alpha}z_0\rangle\right)_+ + \frac{(1+\lambda\alpha)}{2\alpha}\|z - \tfrac{1}{1+\lambda\alpha}z_0\|^2.$$

Applying Lemma B.1 with $y_0 \leftarrow \tfrac{1}{1+\lambda\alpha}z_0$, $c \leftarrow \hat{c}$, $a \leftarrow \mathbf{D}^{-1/2}a$, $\beta \leftarrow \tfrac{\alpha}{1+\lambda\alpha}$ gives

$$z^+ = \frac{1}{1+\lambda\alpha}z_0 - \min\left\{\frac{\alpha}{1+\lambda\alpha}, \frac{(\hat{c})_+}{\|a\|^2_{\mathbf{D}^{-1}}}\right\}\mathbf{D}^{-1/2}a.$$

Changing variables back using $y^+ = \mathbf{D}^{-1/2}z^+$, substituting $\hat{c} = c - \tfrac{\lambda\alpha}{1+\lambda\alpha}\langle a, y_0\rangle$ and re-arranging the above gives

$$y^+ = \frac{1}{1+\lambda\alpha}y_0 - \min\left\{\frac{\alpha}{1+\lambda\alpha}, \frac{\left(c - \tfrac{\lambda\alpha}{1+\lambda\alpha}\langle a, y_0\rangle\right)_+}{\|a\|^2_{\mathbf{D}^{-1}}}\right\}\mathbf{D}^{-1}a$$

$$= \frac{1}{1+\lambda\alpha}\left[y_0 - \min\left\{\alpha, \frac{\left((1+\lambda\alpha)c - \lambda\alpha\langle a, y_0\rangle\right)_+}{\|a\|^2_{\mathbf{D}^{-1}}}\right\}\mathbf{D}^{-1}a\right]. \tag{28}$$

$\square$

## C  Missing Proofs

### C.1  Proof of Lemma 2.1

**Lemma 2.1.** [MoMo update] Let

$$d_k := \sum_{j=1}^{k} \rho_{j,k}\nabla f(x^j, s_j), \quad \bar{f}_k := \sum_{j=1}^{k} \rho_{j,k} f(x^j, s_j), \quad \gamma_k := \sum_{j=1}^{k} \rho_{j,k}\langle \nabla f(x^j, s_j), x^j\rangle. \tag{11}$$

Using model (9), the closed form solution to (10) is

$$x^{k+1} = x^k - \tau_k d_k, \quad \tau_k := \min\left\{\frac{\alpha_k}{\rho_k}, \frac{\left(\bar{f}_k + \langle d_k, x^k\rangle - \gamma_k - \rho_k f_*^k\right)_+}{\|d_k\|^2}\right\}. \tag{12}$$

*Proof.* Recall problem (10) given by

$$x^{k+1} = \operatorname*{argmin}_{y \in \mathbb{R}^d} m_k(y) + \frac{1}{2\alpha_k}\|y - x^k\|^2.$$

Introducing

$$h_k := \sum_{j=1}^{k} \rho_{j,k}[f(x^j, s_j) + \langle \nabla f(x^j, s_j), x^k - x^j\rangle] = \bar{f}_k + \langle d_k, x^k\rangle - \gamma_k, \tag{29}$$

we have that

$$m_k(y) = \max\left\{\rho_k^{-1}(h_k + \langle d_k, y - x^k\rangle), f_*^k\right\} = \left(\rho_k^{-1}(h_k + \langle d_k, y - x^k\rangle) - f_*^k\right)_+ + f_*^k. \tag{30}$$

Using (30), dropping the constant term $f_*^k$, and multiplying with $\rho_k$, problem (10) is equivalent to

$$x^{k+1} = \operatorname*{argmin}_{y \in \mathbb{R}^d} \left( h_k + \langle d_k, y - x^k \rangle - \rho_k f_*^k \right)_+ + \frac{\rho_k}{2\alpha_k} \|y - x^k\|^2.$$

Applying Lemma B.1 with $\beta \leftarrow \rho_k^{-1} \alpha_k$, $c \leftarrow h_k - \rho_k f_*^k$, $a \leftarrow d_k$ and $y_0 \leftarrow x^k$ gives the result. $\qquad \square$

## C.2 Proof of Lemma 3.1

**Lemma 3.1.** Using model (9), the closed form solution to (14) is given by

$$\tau_k = \min\left\{ \frac{\alpha_k}{\rho_k}, \frac{\left((1 + \alpha_k \lambda)(\bar{f}_k - \rho_k f_*^k - \gamma_k) + \langle d_k, x^k \rangle\right)_+}{\|d_k\|_{\mathbf{D}_k^{-1}}^2} \right\}, \tag{15}$$

$$x^{k+1} = \frac{1}{1 + \alpha_k \lambda} \left[ x^k - \tau_k \mathbf{D}_k^{-1} d_k \right]. \tag{16}$$

*Proof.* Recall problem (14) given by

$$x^{k+1} = \operatorname*{argmin}_{y \in \mathbb{R}^d} m_k(y) + \frac{1}{2\alpha_k} \|y - x^k\|_{\mathbf{D}_k}^2 + \frac{\lambda}{2} \|y\|_{\mathbf{D}_k}^2.$$

We use again (30). Dropping the constant term $f_*^k$, and multiplying with $\rho_k$, problem (14) is equivalent to

$$x^{k+1} = \operatorname*{argmin}_{y \in \mathbb{R}^d} \left( h_k + \langle d_k, y - x^k \rangle - \rho_k f_*^k \right)_+ + \frac{\rho_k}{2\alpha_k} \|y - x^k\|_{\mathbf{D}_k}^2 + \frac{\rho_k \lambda}{2} \|y\|_{\mathbf{D}_k}^2.$$

Now applying Lemma B.2 with $y_0 \leftarrow x^k$, $a \leftarrow d_k$, $c \leftarrow h_k - \rho_k f_*^k$, $\lambda \leftarrow \rho_k \lambda$, $\alpha \leftarrow \rho_k^{-1} \alpha_k$ and $\mathbf{D} \leftarrow \mathbf{D}_k$, we obtain the result. $\qquad \square$

# D Estimating a Lower Bound: Proofs and Alternatives

## D.1 Proof of Lemma 4.2

**Lemma 4.2.** Let $f(x, s)$ be convex in $x$ for all $s \in \mathcal{D}$. Let $x^k$ be given by (16) with $\lambda = 0$. Let $\eta_k := \prod_{j=2}^k \lambda_{\min}(\mathbf{D}_j^{-1} \mathbf{D}_{j-1})$, and $h_k := \bar{f}_k + \langle d_k, x^k \rangle - \gamma_k$. We have $\bar{f}_*^k \geq f_*^{k+1}$ where

$$f_*^{k+1} := \frac{1}{2\eta_k \tau_k \rho_k} \left( \sum_{j=1}^k 2\eta_j \tau_j \left( h_j - \tfrac{1}{2} \tau_j \|d_j\|_{\mathbf{D}_j^{-1}}^2 \right) - D_1^2 - 2\sum_{j=1}^{k-1} \eta_j \tau_j \rho_j \bar{f}_*^j \right)$$

where $D_1 := \|x^1 - x^*\|_{\mathbf{D}_1}$. Bootstrapping by using $f_*^k \approx \bar{f}_*^{k-1}$ we have for $k \geq 2$ that

$$f_*^{k+1} = \frac{1}{\rho_k} \left( h_k - \tfrac{1}{2} \tau_k \|d_k\|_{\mathbf{D}_k^{-1}}^2 \right). \tag{19}$$

*Proof.* Consider the update (16) without weight decay, that is $\lambda = 0$, and switching the index $k \to j$, which is

$$x^{j+1} = x^j - \tau_j \mathbf{D}_j^{-1} d_j,$$

where $\tau_j$ is the step size. Subtracting $x^*$ from both sides, taking norms and expanding the squares we have that

$$\left\| x^{j+1} - x^* \right\|_{\mathbf{D}_j}^2 = \left\| x^j - x^* \right\|_{\mathbf{D}_j}^2 - 2\tau_j \langle d_j, x^j - x^* \rangle + \tau_j^2 \|d_j\|_{\mathbf{D}_j^{-1}}^2. \tag{31}$$

Now let $\delta_{j+1} := \lambda_{\min}(\mathbf{D}_{j+1}^{-1} \mathbf{D}_j)$ and note that for every vector $v \in \mathbb{R}^d$ we have that

$$\delta_{j+1} \|v\|_{\mathbf{D}_{j+1}}^2 \leq \|v\|_{\mathbf{D}_j}^2. \tag{32}$$

Indeed this follows since

$$\|v\|_{\mathbf{D}_j}^2 = v^\top \mathbf{D}_j v = v^\top \mathbf{D}_{j+1}^{1/2}\big(\mathbf{D}_{j+1}^{-1/2}\mathbf{D}_j\mathbf{D}_{j+1}^{-1/2}\big)\mathbf{D}_{j+1}^{1/2}v$$
$$\geq \lambda_{\min}\big(\mathbf{D}_{j+1}^{-1}\mathbf{D}_j\big)\|v\|_{\mathbf{D}_{j+1}}^2 = \delta_{j+1}\|v\|_{\mathbf{D}_{j+1}}^2.$$

For simplicity, denote $\nabla f_l = \nabla f(x^l, s_l), f_l = f(x^l, s_l)$. We have that

$$\big\langle d_j, x^j - x^* \big\rangle = \sum_{l=1}^{j} \rho_{l,j}\big\langle \nabla f_l, x^j - x^* \big\rangle$$

$$= \sum_{l=1}^{j} \rho_{l,j}\big(\big\langle \nabla f_l, x^j - x^l \big\rangle + \big\langle \nabla f_l, x^l - x^* \big\rangle\big)$$

$$\geq \sum_{l=1}^{j} \rho_{l,j}\big(\big\langle \nabla f_l, x^j - x^l \big\rangle + f_l - f(x^*, s_l)\big) \qquad \text{(by convexity of } f(\cdot, s))$$

$$= \bar{f}_j + \big\langle d_j, x^j \big\rangle - \gamma_j - \sum_{l=1}^{j} \rho_{l,j} f(x^*, s_l) \;=\; h_j - \rho_j \bar{f}_*^j. \qquad (33)$$

Using (32) together with (33) in (31) gives

$$\delta_{j+1}\big\|x^{j+1} - x^*\big\|_{\mathbf{D}_{j+1}}^2 \leq \big\|x^{j+1} - x^*\big\|_{\mathbf{D}_j}^2$$

$$= \big\|x^j - x^*\big\|_{\mathbf{D}_j}^2 - 2\tau_j\big\langle d_j, x^j - x^* \big\rangle + \tau_j^2\|d_j\|_{\mathbf{D}_j^{-1}}^2$$

$$\leq \big\|x^j - x^*\big\|_{\mathbf{D}_j}^2 - 2\tau_j(h_j - \rho_j\bar{f}_*^j) + \tau_j^2\|d_j\|_{\mathbf{D}_j^{-1}}^2. \qquad (34)$$

Now we will perform a weighted telescoping. We will multiply the above by $\eta_j > 0$ such that $\delta_{j+1}\eta_j = \eta_{j+1}$, thus $\eta_j = \eta_1\prod_{l=2}^{j}\delta_l$. Thus multiplying through by $\eta_j$ we have that

$$\eta_{j+1}\big\|x^{j+1} - x^*\big\|_{\mathbf{D}_{j+1}}^2 \leq \eta_j\big\|x^j - x^*\big\|_{\mathbf{D}_j}^2 - 2\eta_j\tau_j(h_j - \rho_j\bar{f}_*^j) + \eta_j\tau_j^2\|d_j\|_{\mathbf{D}_j^{-1}}^2.$$

Summing up from $j = 1, \ldots, k$ and telescoping we have that

$$0 \leq \eta_{k+1}\big\|x^{k+1} - x^*\big\|_{\mathbf{D}_{k+1}}^2$$

$$\leq \eta_1\big\|x^1 - x^*\big\|_{\mathbf{D}_1}^2 - 2\sum_{j=1}^{k}\eta_j\tau_j(h_j - \rho_j\bar{f}_*^j) + \sum_{j=1}^{k}\eta_j\tau_j^2\|d_j\|_{\mathbf{D}_j^{-1}}^2. \qquad (35)$$

Re-arranging the above, choosing $\eta_1 = 1$ and isolating $\bar{f}_*^k$ gives

$$2\eta_k\tau_k\rho_k\bar{f}_*^k \geq 2\sum_{j=1}^{k}\eta_j\tau_j h_j - \big\|x^1 - x^*\big\|_{\mathbf{D}_1}^2 - \sum_{j=1}^{k}\eta_j\tau_j^2\|d_j\|_{\mathbf{D}_j^{-1}}^2 - 2\sum_{j=1}^{k-1}\eta_j\tau_j\rho_j\bar{f}_*^j.$$

Dividing through by $2\eta_k\tau_k\rho_k$ gives the main result. Finally the recurrence follows since, for $k \geq 2$ we have that

$$f_*^{k+1} := \frac{2\sum_{j=1}^{k}\eta_j\tau_j h_j - \big\|x^1 - x^*\big\|_{\mathbf{D}_1}^2 - \sum_{j=1}^{k}\eta_j\tau_j^2\|d_j\|_{\mathbf{D}_j^{-1}}^2 - 2\sum_{j=1}^{k-1}\eta_j\tau_j\rho_j\bar{f}_*^j}{2\eta_k\tau_k\rho_k}$$

$$= \frac{\eta_{k-1}\tau_{k-1}\rho_{k-1}}{\eta_k\tau_k\rho_k}\underbrace{\frac{2\sum_{j=1}^{k-1}\eta_j\tau_j h_j - \big\|x^1 - x^*\big\|_{\mathbf{D}_1}^2 - \sum_{j=1}^{k-1}\eta_j\tau_j^2\|d_j\|_{\mathbf{D}_j^{-1}}^2 - 2\sum_{j=1}^{k-2}\eta_j\tau_j\rho_j\bar{f}_*^j}{2\eta_{k-1}\tau_{k-1}\rho_{k-1}}}_{=f_*^k}$$

$$+ \frac{\eta_{k-1}\tau_{k-1}\rho_{k-1}}{\eta_k\tau_k\rho_k}\frac{2\eta_k\tau_k h_k - \eta_k\tau_k^2\|d_k\|_{\mathbf{D}_k^{-1}}^2 - 2\eta_{k-1}\tau_{k-1}\rho_{k-1}\bar{f}_*^{k-1}}{2\eta_{k-1}\tau_{k-1}\rho_{k-1}}$$

$$= \frac{2\eta_{k-1}\tau_{k-1}\rho_{k-1}(f_*^k - \bar{f}_*^{k-1}) - \eta_k\tau_k^2\|d_k\|_{\mathbf{D}_k^{-1}}^2 + 2\eta_k\tau_k h_k}{2\eta_k\tau_k\rho_k}.$$

Now bootstrapping by using $f_*^k \approx \bar{f}_*^{k-1}$ gives the result. $\qquad \square$

## D.2 The Max Lower Bound

Here we derive an alternative estimate for the lower bound that does not require bootstrapping, contrary to Lemma 4.2.

**Lemma D.1.** Let $f(x,s)$ be convex in $x$ for every sample $s$. Furthermore let $x^* \in \underset{x \in \mathbb{R}^d}{\operatorname{argmin}} f(x)$. Consider $x^k$ are the iterates of (16) with $\lambda = 0$ and let

$$\eta_k = \prod_{j=2}^{k} \lambda_{\min}\left(\mathbf{D}_j^{-1}\mathbf{D}_{j-1}\right), \; \bar{f}_*^k := \frac{1}{\rho_k}\sum_{j=1}^{k}\rho_{j,k}f(x^*, s_j), \; h_k = \bar{f}_k + \langle d_k, x^k \rangle - \gamma_k.$$

It follows that

$$\max_{j=1,\ldots,k} \bar{f}_*^j \geq f_*^{k+1} := \frac{2\sum_{j=1}^{k}\eta_j\tau_j h_j - \left\|x^1 - x^*\right\|^2 - \sum_{j=1}^{k}\eta_j\tau_j^2 \|d_j\|_{\mathbf{D}_j^{-1}}^2}{2\sum_{j=1}^{k}\eta_j\tau_j\rho_j}. \tag{36}$$

Furthermore we have the recurrence

$$f_*^{k+1} = \frac{f_*^k \sum_{j=1}^{k-1}\eta_j\tau_j\rho_j + \eta_k\tau_k\left(h_k - \frac{1}{2}\tau_k \|d_k\|_{\mathbf{D}_k^{-1}}^2\right)}{\sum_{j=1}^{k}\eta_j\tau_j\rho_j}. \tag{37}$$

In particular when $\mathbf{D}_k = \mathbf{Id}$ for every $k$, then we have that $\eta_k = 1$ for all $k$.

*Proof.* From step (35) and re-arranging we have that

$$2\left(\max_{j=1,\ldots,k}\bar{f}_*^j\right)\left(\sum_{j=1}^{k}\eta_j\tau_j\rho_j\right) \geq 2\left(\sum_{j=1}^{k}\eta_j\tau_j\rho_j\right)\bar{f}_*^j$$

$$\geq 2\sum_{j=1}^{k}\eta_j\tau_j h_j - \left\|x^1 - x^*\right\|_{\mathbf{D}_1}^2 - \sum_{j=1}^{k}\eta_j\tau_j^2 \|d_j\|_{\mathbf{D}_j^{-1}}^2.$$

If we now assume that $\bar{f}_*^j \approx f(x^*)$ (or upper bounding $\bar{f}_*^j$ by a constant) then by substituting in $f(x^*)$, dividing through by $\left(\sum_{j=1}^{k}\eta_j\tau_j\rho_j\right)$ gives the estimate

$$\max_{j=1,\ldots,k}\bar{f}_*^j \geq f_*^{k+1} := \frac{2\sum_{j=1}^{k}\eta_j\tau_j h_j - \left\|x^1 - x^*\right\|^2 - \sum_{j=1}^{k}\eta_j\tau_j^2 \|d_j\|_{\mathbf{D}_j^{-1}}^2}{2\sum_{j=1}^{k}\eta_j\tau_j\rho_j}.$$

Finally the recurrence follows since

$$f_*^{k+1} = \frac{2\sum_{j=1}^{k}\eta_j\tau_j h_j - \left\|x^1 - x^*\right\|_{\mathbf{D}_1}^2 - \sum_{j=1}^{k}\eta_j\tau_j^2 \|d_j\|_{\mathbf{D}_j^{-1}}^2}{2\sum_{j=1}^{k}\eta_j\tau_j\rho_j}$$

$$= \frac{\sum_{j=1}^{k-1}\eta_j\tau_j\rho_j}{\sum_{j=1}^{k}\eta_j\tau_j\rho_j}\frac{2\sum_{j=1}^{k-1}\eta_j\tau_j h_j - \left\|x^1 - x^*\right\|_{\mathbf{D}_1}^2 - \sum_{j=1}^{k-1}\eta_j\tau_j^2 \|d_j\|_{\mathbf{D}_j^{-1}}^2}{2\sum_{j=1}^{k-1}\eta_j\tau_j\rho_j}$$

$$\quad + \frac{2\eta_k\tau_k h_k - \eta_k\tau_k^2 \|d_k\|_{\mathbf{D}_k^{-1}}^2}{2\sum_{j=1}^{k}\eta_j\tau_j\rho_j}$$

$$= \frac{f_*^k \sum_{j=1}^{k-1}\eta_j\tau_j\rho_j + \eta_k\tau_k\left(h_k - \frac{1}{2}\tau_k \|d_k\|_{\mathbf{D}_k^{-1}}^2\right)}{\sum_{j=1}^{k}\eta_j\tau_j\rho_j}.$$

$\square$

# E  ADDITIONAL INFORMATION ON EXPERIMENTS

## E.1  ADDITIONAL PLOTS

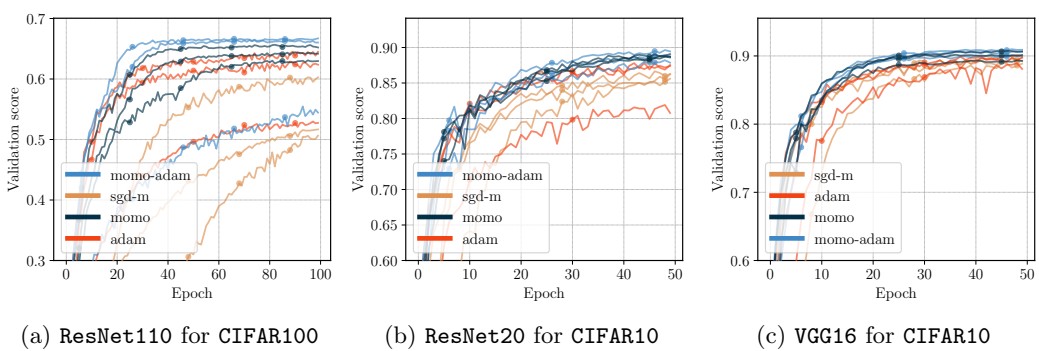

(a) `ResNet110` for `CIFAR100`    (b) `ResNet20` for `CIFAR10`    (c) `VGG16` for `CIFAR10`

Figure E.1: Validation score over training, we plot, for each method, the three choices of $\alpha_0$ that lead to the best validation score (compare to Fig. 2).

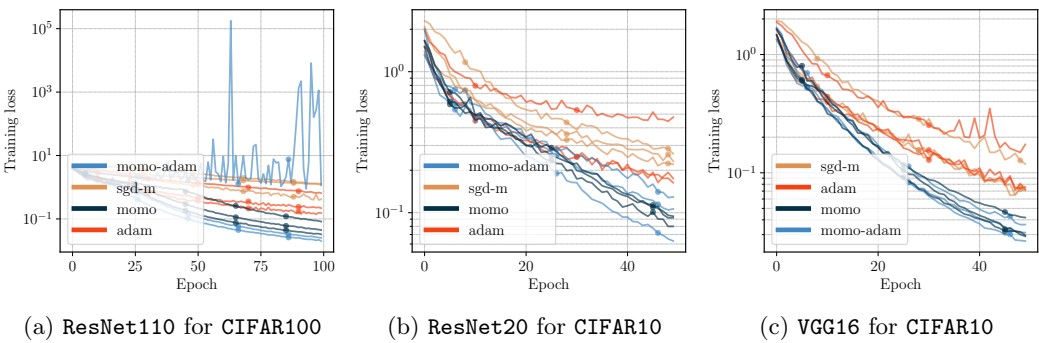

(a) `ResNet110` for `CIFAR100`    (b) `ResNet20` for `CIFAR10`    (c) `VGG16` for `CIFAR10`

Figure E.2: Training loss over training, we plot, for each method, the three choices of $\alpha_0$ that lead to the best validation score.

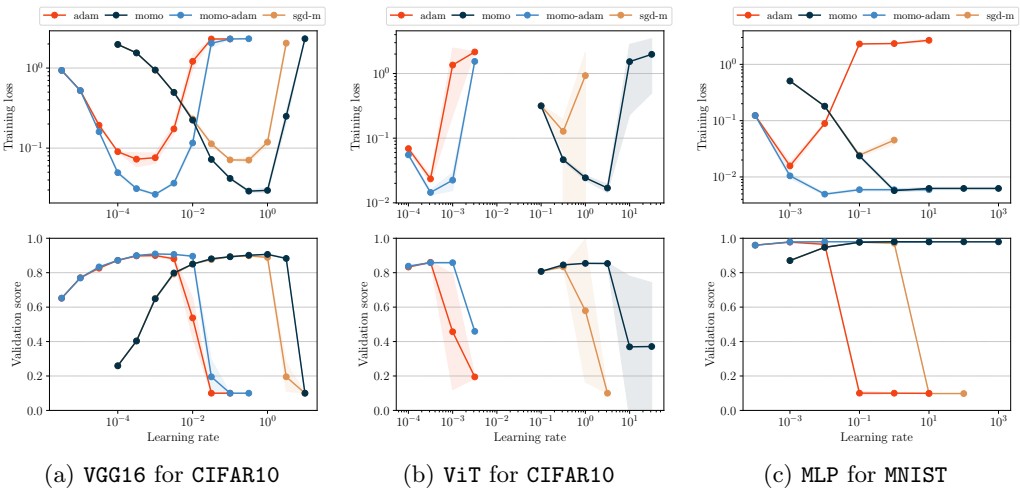

(a) `VGG16` for `CIFAR10`    (b) `ViT` for `CIFAR10`    (c) `MLP` for `MNIST`

Figure E.3: Training loss (top row) and validation accuracy (bottom row) after a fixed number of epochs, for varying (constant) learning rate $\alpha_0$.

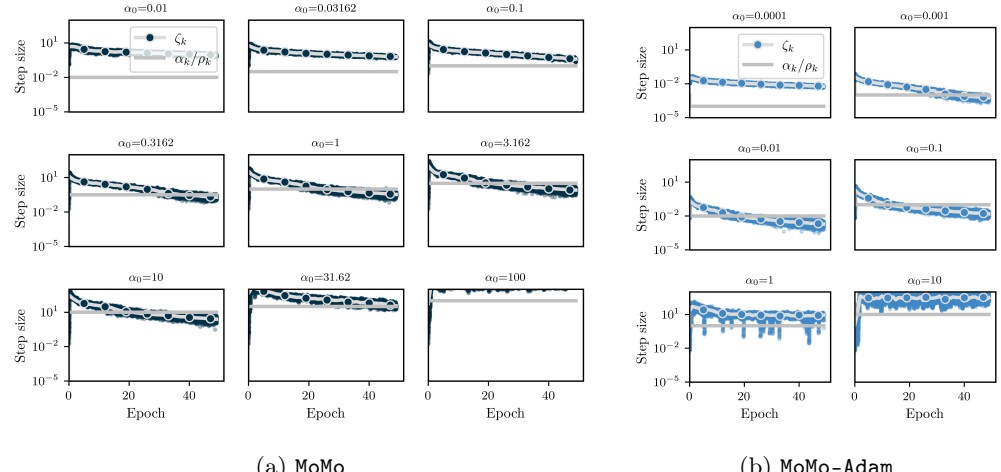

(a) `MoMo`  (b) `MoMo-Adam`

Figure E.4: `ResNet20` for `CIFAR10`. Adaptive learning rate of `MoMo` (left) and `MoMo-Adam` (right). The colored dots represent the term $\zeta_k$ in each iteration. The grey line represents the user-specified learning rate $\alpha_k/\rho_k$ (note that $\rho_k = 1$ for `MoMo` and $\rho_k \approx 1$ except for the first few iterations in `MoMo-Adam`). The minimum of the grey line and the dots is the adaptive learning rate $\tau_k = \min\{\frac{\alpha_k}{\rho_k}, \zeta_k\}$ in each iteration. The silver line with colored markers is the median over the values of $\zeta_k$ in each epoch.

|  | MoMo | MoMo-Adam | SGD-M | Adam |
|---|---|---|---|---|
| ResNet110 for CIFAR100 | 65.21 $\pm 1.61$ | **66.71** $\pm 0.31$ | 60.28 $\pm 0.36$ | 64.5 $\pm 1.14$ |
| ResNet20 for CIFAR10 | 89.07 $\pm 0.2$ | **89.45** $\pm 0.17$ | 86.27 $\pm 0.67$ | 87.54 $\pm 0.26$ |
| ViT for CIFAR10 | 85.43 $\pm 0.19$ | 85.81 $\pm 0.57$ | 83.39 $\pm 0.28$ | 86.02 $\pm 0.44$ |
| VGG16 for CIFAR10 | 90.64 $\pm 0.18$ | **90.9** $\pm 0.17$ | 89.81 $\pm 0.43$ | 89.95 $\pm 0.67$ |
| MLP for MNIST | **97.97** $\pm 0.08$ | 97.96 $\pm 0.12$ | 97.73 $\pm 0.12$ | 97.75 $\pm 0.06$ |
| DLRM for Criteo | 78.83 $\pm 0.038$ | 78.98 $\pm 0.036$ | 78.81 $\pm 0.041$ | **79.05** $\pm 0.014$ |
| ResNet18 for Imagenet32 | **47.66**[*] | 47.54[*] | 47.38 | 46.98 |
| ResNet18 for Imagenet-1k | **69.68** | N/A | 69.57 | N/A |
| IWSLT14 (dp 0.1) | N/A | **33.63**[*] | N/A | 32.56 |
| IWSLT14 (dp 0.3) | N/A | **35.34**[*] | N/A | 34.97 |

Table 1: Validation score (with one standard deviation) for the best learning rate choice for each method among the ones displayed in Section 5. Symbol "*" indicates usage of online lower bound, otherwise `MoMo(-Adam)` used with $f_\star^k = 0$. Bold indicates the best method (for experiments with multiple seeds, we only mark in bold if the advantage is outside of standard deviation).

## E.2 Experimental Setup of Section 5.1

We set the momentum parameter $\beta = 0.9$ for `MoMo` and `SGD-M`, and $(\beta_1, \beta_2) = (0.9, 0.999)$ for `MoMo-Adam` and `Adam` respectively. We do not use weight decay, i.e. $\lambda = 0$.

For `SGD-M` we set the dampening parameter (in `Pytorch`) equal to the momentum parameter 0.9. Like this, `SGD-M` does an exponentially-weighted average of past gradients and hence is comparable to `MoMo` for identical learning rate and momentum. Setting `dampening` $= 0.9$ is equivalent to running with `dampening` $= 0$ and a ten times smaller learning rate. For all other hyperparameters we use the `Pytorch` default values for `Adam` and `SGD-M` (unless explicitly stated otherwise).

## E.3 Models and Datasets

**ResNet for CIFAR**                                                    (He et al., 2016)

Used for `ResNet20` for `CIFAR10` and `ResNet110` for `CIFAR100`. We adapt the last layer

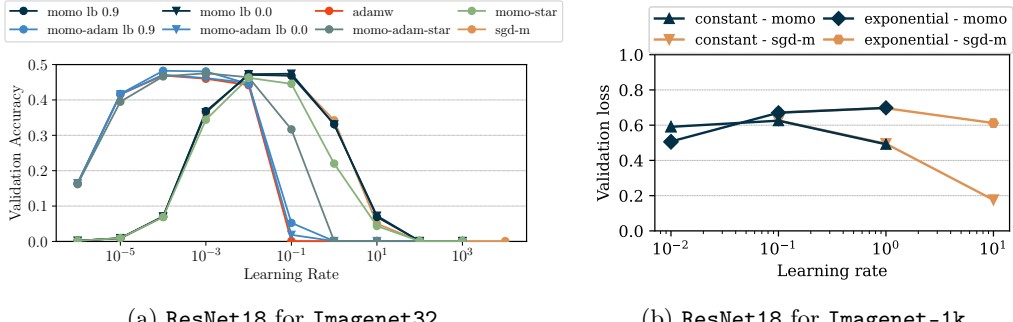

(a) `ResNet18` for `Imagenet32`         (b) `ResNet18` for `Imagenet-1k`

Figure E.5: Left: Validation accuracy of a `ResNet18` for `Imagenet32` with weight decay $\lambda = 10^{-4}$. Right: Validation accuracy of a `ResNet18` for `Imagenet-1k`, with standard exponential learning rate schedule (decay factor 10 at epochs 30 and 60) and constant learning rate schedule.

output size to $\{10, 100\}$ according to the used dataset. We run 50 epochs for `ResNet20` and 100 epochs for `ResNet110`.

Model     https://github.com/akamaster/pytorch_resnet_cifar10/blob/master/resnet.py

**VGG16 for CIFAR10**                                                    (Simonyan & Zisserman, 2015)

A deep network with 16 convolutional layers. We run 50 epochs.

Model     https://github.com/chengyangfu/pytorch-vgg-cifar10/blob/master/vgg.py

**ViT for CIFAR10**                                                          (Dosovitskiy et al., 2021)

A small vision transformer, based on the hyperparameter setting proposed in github.com/ kentaroy47/vision-transformers-cifar10. In particular, we set the patch size to four. We run 200 epochs.

Model     https://github.com/lucidrains/vit-pytorch

**ResNet18 for Imagenet32**                                                       (He et al., 2016)

`Imagenet32` is a downsampled version of `Imagenet-1k` to images of $32 \times 32$ pixels. We adapt the last layer output size to 1000. We run 45 epochs.

Model     https://github.com/kuangliu/pytorch-cifar/blob/master/models/resnet.py

**ResNet18 for Imagenet-1k**                                                      (He et al., 2016)

We use both a constant learning rate and a schedule that decays the learning rate by 0.1 every 30 epochs. We run 90 epochs. Note that for `SGD-M` the decaying schedule with initial learning rate of 0.1 is considered state-of-the-art. As we set `dampening` $= 0.9$, and this is equivalent to `dampening` $= 0$ and a ten times smaller learning rate (see Appendix E.2), in our plots the best score is displayed for initial learning rate of 1 accordingly.

Model     pytorch.org/vision/main/models/generated/torchvision.models.resnet18.html

**DLRM for Criteo**                                                        (Jean-Baptiste Tien, 2014)

`DLRM` is an industry-scale model with over 300 million parameters. the `Criteo` dataset contains approximately 46 million training samples. We run 300k iterations with batch size 128.

Dataset     https://kaggle.com/c/criteo-display-ad-challenge
Model       https://github.com/facebookresearch/dlrm

**IWSLT14**                                                                    (Ott et al., 2019)

We use a transformer with six encoder and decoder blocks from `fairseq`. The training loss is the cross-entropy loss with label smoothing of 0.1. We use weight decay of $\lambda = 10^{-4}$ (although we noticed that weight decay does not influence the performance of `MoMo-Adam`), momentum parameters $(\beta_1, \beta_2) = (0.9, 0.98)$. We train for 60 epochs.

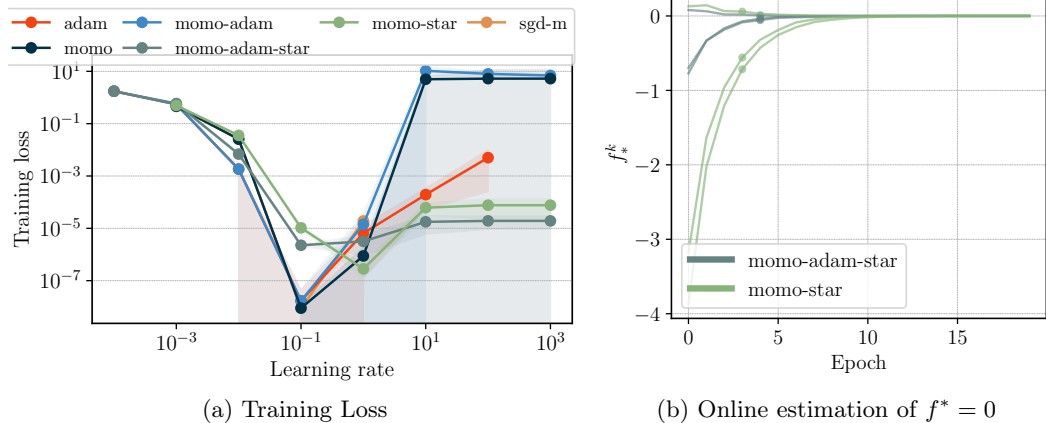

(a) Training Loss

(b) Online estimation of $f^* = 0$

Figure E.6: Illustrative example of online lower bound estimation. For all MoMo methods, we initialize $f_*^1 = -10$. Left: Training loss for varying (constant) learning rate $\alpha_0$. Right: Value of $f_*^k$ over training, one line corresponds to one choice of $\alpha_0$. We plot per method the four values of $\alpha_0$ that lead to smallest training loss.

Model    https://github.com/facebookresearch/fairseq

For each experiments, we list how long one training run approximately takes on the hardware we use. Unless specified otherwise, we train on a single NVIDIA A100 GPU. `ResNet110` for `CIFAR100` 90 min, `ResNet20` for `CIFAR10` 30 min, `VGG16` for `CIFAR10` 30 min, `MLP` for `MNIST` 3 min, `ResNet18` for `Imagenet32` 20 hours (on NVIDIA V100), Transformer for `IWSLT14` 3 hours.

### E.4    Illustrative Example of Online Lower Bound Estimation

We show how our online estimation of $f_*^k$, derived in Section 4 and Lemma 4.2, work for a simple example. Consider a regression problem, with synthetic matrix $A \in \mathbb{R}^{200 \times 10}$ and $b \in \mathbb{R}^{200}$. We solve the problem $\min_{x \in \mathbb{R}^{10}} \sum_{i=1}^{200} \frac{1}{2} \|a_i^\top x - b_i\|^2$, where $a_i$ are the rows of $A$. The data is generated in a way such that there exists $\hat{x}$ with $b = A\hat{x}$ and hence the optimal value is $f^* = 0$.

We now run MoMo(-Adam) with lower bound estimate $f_*^k = -10$ in all iterations, and MoMo(-Adam)* with initialization $f_*^1 = -10$. Clearly, this is not a tight estimate of the optimal value $f^*$. From Fig. E.6a, we see that online estimation of $f_*^k$, used in MoMo(-Adam)*, improves stability of the training compared to plain MoMo(-Adam) where a constant value $f_*^k = -10$ is used. From Fig. E.6b, we also see that the online values of $f_*^k$ converge to $f^* = 0$.

## F    Convergence Analysis

Here we give another motivation for a variant of MoMo through convexity. We discovered this interpretation of MoMo after reading the concurrent work (Wang et al., 2023).

For this alternative derivation of MoMo, first let $\tau_k \geq 0$ be a free parameter, and consider a general momentum method with a preconditioner given by

$$d_k = \sum_{j=1}^{k} \rho_{j,k} \nabla f(x^j, s_j),$$
$$x^{k+1} = x^k - \tau_k \mathbf{D}_k^{-1} d_k. \tag{38}$$

We can now view $x^{k+1}$ as a function of $\tau_k$, that is $x^{k+1}(\tau_k)$. Ideally we would like to choose $\tau_k$ so that $x^{k+1}$ is as close as possible to the optimum solution $x^*$, that is to minimize

$\left\|x^{k+1}(\tau_k) - x^*\right\|^2_{\mathbf{D}_k}$ in $\tau_k$. This is general not possible because we do not know $x^*$. But if we assume that $f(\cdot, s)$ is a convex function, then we can minimize an upper bound of $\left\|x^{k+1}(\tau_k) - x^*\right\|^2_{\mathbf{D}_k}$ with respect to $\tau_k$. As we show next, this gives the adaptive term in the learning rate of MoMo if $f^k_* = \bar{f}^k_*$.

**Lemma F.1.** Let $f(\cdot, s)$ be convex for every $s$. Let $h_k := \bar{f}_k + \langle d_k, x^k \rangle - \gamma_k$ where $d_k, \bar{f}_k$, and $\gamma_k$ are defined in (11). Consider the iterates given by (38) and let $x^* \in \arg\min_{x \in \mathbb{R}^d} f(x)$. Then, we have the upper bound

$$\left\|x^{k+1} - x^*\right\|^2_{\mathbf{D}_k} \leq \left\|x^k - x^*\right\|^2_{\mathbf{D}_k} - 2\tau_k(h_k - \rho_k \bar{f}^k_*) + \tau_k^2 \|d_k\|^2_{\mathbf{D}_k^{-1}}. \tag{39}$$

The minimum of the right-hand side of (39), over the set $\tau_k \in \mathbb{R}_{\geq 0}$, is attained at

$$\bar{\tau}_k = \frac{(h_k - \rho_k \bar{f}^k_*)_+}{\|d_k\|^2_{\mathbf{D}_k^{-1}}}. \tag{40}$$

*Proof.* Subtracting $x^*$ from both sides, taking norms and expanding the squares gives

$$\left\|x^{k+1} - x^*\right\|^2_{\mathbf{D}_k} = \left\|x^k - x^*\right\|^2_{\mathbf{D}_k} - 2\tau_k \langle d_k, x^k - x^* \rangle + \tau_k^2 \|d_k\|^2_{\mathbf{D}_k^{-1}}. \tag{41}$$

Denote $\nabla f_j := \nabla f(x^j, s_j)$, $f_j := f(x^j, s_j)$. Now using that

$$\begin{aligned}
\langle d_k, x^k - x^* \rangle &= \sum_{j=1}^k \rho_{j,k} \langle \nabla f_j, x^k - x^* \rangle \\
&= \sum_{j=1}^k \rho_{j,k} \left( \langle \nabla f_j, x^k - x^j \rangle + \langle \nabla f_j, x^j - x^* \rangle \right) \\
&\geq \sum_{j=1}^k \rho_{j,k} \left( \langle \nabla f_j, x^k - x^j \rangle + f_j - f(x^*, s_j) \right) \qquad \text{(by convexity of } f(\cdot, s_j)) \\
&= \langle d_k, x^k \rangle - \gamma_k + \sum_{j=1}^k \rho_{j,k}(f_j - f(x^*, s_j)) = h_k - \rho_k \bar{f}^k_*. \tag{42}
\end{aligned}$$

Using (42) in (41) gives

$$\begin{aligned}
\left\|x^{k+1} - x^*\right\|^2_{\mathbf{D}_k} &= \left\|x^k - x^*\right\|^2_{\mathbf{D}_k} - 2\tau_k \langle d_k, x^k - x^* \rangle + \tau_k^2 \|d_k\|^2_{\mathbf{D}_k^{-1}} \\
&\leq \left\|x^k - x^*\right\|^2_{\mathbf{D}_k} - 2\tau_k(h_k - \rho_k \bar{f}^k_*) + \tau_k^2 \|d_k\|^2_{\mathbf{D}_k^{-1}}.
\end{aligned}$$

If we now minimize the right-hand side of the above in $\tau_k$, but restricted to $\tau_k \geq 0$, we arrive at (40). $\square$

Inequality (39) holds for any choice of $\tau_k \geq 0$ in (38), in particular for $\tau_k = \min\{\frac{\alpha_k}{\rho_k}, \frac{(h_k - \rho_k \bar{f}^k_*)_+}{\|d_k\|^2_{\mathbf{D}_k^{-1}}}\}$. This choice for $\tau_k$ is equal to MoMo for $\lambda = 0$ and $f^k_* = \bar{f}^k_*$. As a consequence, we we can prove a descent lemma for MoMo.

**Lemma 4.1.** Let $f(\cdot, s)$ be convex for every $s$ and let $x^* \in \arg\min_{x \in \mathbb{R}^d} f(x)$. For the iterates of the general MoMo update (cf. Lemma 3.1) with $\lambda = 0$ and $f^k_* = \bar{f}^k_*$, it holds

$$\left\|x^{k+1} - x^*\right\|^2_{\mathbf{D}_k} \leq \left\|x^k - x^*\right\|^2_{\mathbf{D}_k} - \tau_k(h_k - \rho_k \bar{f}^k_*)_+. \tag{18}$$

*Proof.* We again denote $h_k = \bar{f}_k + \langle d_k, x^k \rangle - \gamma_k$. First, assume $\tau_k = \frac{(h_k - \rho_k \bar{f}_*^k)_+}{\|d_k\|_{\mathbf{D}_k^{-1}}^2}$. Inserting this $\tau_k$ back in (39) we have that

$$
\begin{aligned}
\|x^{k+1} - x^*\|_{\mathbf{D}_k}^2 &\leq \|x^k - x^*\|_{\mathbf{D}_k}^2 - 2\frac{(h_k - \rho_k \bar{f}_*^k)_+}{\|d_k\|_{\mathbf{D}_k^{-1}}^2}(h_k - \rho_k \bar{f}_*^k) + \frac{(h_k - \rho_k \bar{f}_*^k)_+^2}{\|d_k\|_{\mathbf{D}_k^{-1}}^2} \\
&= \|x^k - x^*\|_{\mathbf{D}_k}^2 - \frac{(h_k - \rho_k \bar{f}_*^k)_+^2}{\|d_k\|_{\mathbf{D}_k^{-1}}^2} \\
&= \|x^k - x^*\|_{\mathbf{D}_k}^2 - \tau_k(h_k - \rho_k \bar{f}_*^k)_+ .
\end{aligned}
\tag{43}
$$

Here we used that $a(a)_+ = (a)_+^2$ for all $a \in \mathbb{R}$.

If we have $\tau_k = \frac{\alpha_k}{\rho_k}$, then from (39) we get

$$
\|x^{k+1} - x^*\|_{\mathbf{D}_k}^2 \leq \|x^k - x^*\|_{\mathbf{D}_k}^2 + \frac{\alpha_k}{\rho_k}\big[-2(h_k - \rho_k \bar{f}_*^k) + \frac{\alpha_k}{\rho_k}\|d_k\|_{\mathbf{D}_k^{-1}}^2\big].
\tag{44}
$$

Using that in this case $\frac{\alpha_k}{\rho_k} \leq \frac{(h_k - \rho_k \bar{f}_*^k)_+}{\|d_k\|_{\mathbf{D}_k^{-1}}^2}$ and hence $\frac{\alpha_k}{\rho_k}\|d_k\|_{\mathbf{D}_k^{-1}}^2 \leq (h_k - \rho_k \bar{f}_*^k)_+$. Further, it must hold $(h_k - \rho_k \bar{f}_*^k) = (h_k - \rho_k \bar{f}_*^k)_+$ as $\alpha_k > 0$. We get

$$
\begin{aligned}
\|x^{k+1} - x^*\|_{\mathbf{D}_k}^2 &\leq \|x^k - x^*\|_{\mathbf{D}_k}^2 - \frac{\alpha_k}{\rho_k}(h_k - \rho_k \bar{f}_*^k)_+ \\
&= \|x^k - x^*\|_{\mathbf{D}_k}^2 - \tau_k(h_k - \rho_k \bar{f}_*^k)_+ \quad (\tau_k = \frac{\alpha_k}{\rho_k}).
\end{aligned}
\tag{45}
$$

Now, if $\tau_k = \min\{\frac{\alpha_k}{\rho_k}, \frac{(h_k - \rho_k \bar{f}_*^k)_+}{\|d_k\|_{\mathbf{D}_k^{-1}}^2}\}$, either (43) or (45) is true, and hence we have

$$
\|x^{k+1} - x^*\|_{\mathbf{D}_k}^2 \leq \|x^k - x^*\|_{\mathbf{D}_k}^2 - \tau_k(h_k - \rho_k \bar{f}_*^k)_+ .
$$

$\square$

We will need the following interpolation assumption:

$$
f(x^*, s) = \inf_x f(x, s) = f^* \quad \text{for all } s \in \mathcal{D}.
\tag{46}
$$

The following theorem proves convergence of MoMo (Algorithm 1) with $\alpha_k = +\infty$ under interpolation, when the loss functions are convex, and the gradients are either *locally bounded* or the gradients are continuous. This is an unusual result, since in the non-smooth setting, one needs to assume the gradients or the iterates are globally bounded (Orabona, 2019; Garrigos & Gower, 2023), or in the smooth setting (where the gradient is continuous) one needs to assume globally Lipschitz gradients. Here we do not need these assumptions, and instead, rely on interpolation.

**Theorem F.2.** Let $f(\cdot, s)$ be convex for every $s$ and let $x^* \in \arg\min_{x \in \mathbb{R}^d} f(x)$. Assume that (46) holds. Let $(x^k)$ be the iterates of Algorithm 1 with $f_*^k = f^*$, $\alpha_k = +\infty$ for all $k \in \mathbb{N}$ and assume that $d_k \neq 0$ for all $k \in \mathbb{N}$. Define

$$
B := \{x \mid \|x - x^*\| < \|x^1 - x^*\|\}.
$$

Assume that $G^2 := \max_{x \in B} \mathbb{E}\left[\|\nabla f(x, s)\|^2\right] < \infty^a$. Then, it holds

$$
\min_{k=1,\dots,K} \mathbb{E}\left[f(x^k) - f^*\right] \leq \frac{G\|x^1 - x^*\|}{\sqrt{K}(1 - \beta)}.
$$

---
[a]Because $B$ is bounded, this is always satisfied if $\mathcal{D}$ is finite.

*Proof.* Recall that for Algorithm 1 it holds that $\rho_k = 1$, $\mathbf{D}_k = \mathbf{Id}$ in Lemma 3.1. The key quantity is $h_k := \bar{f}_k + \langle d_k, x^k \rangle - \gamma_k$. Let us denote $g_k = \nabla f(x^k, s_k)$. Further, denote with $\mathcal{F}_k$ the $\sigma$-algebra generated by $\{s_1, \dots, s_{k-1}\}$.

**Step 1.** We first show by induction that $h_k - f^* \geq 0$ for all $k \in \mathbb{N}$. For $k = 0$ we have $h_0 = f(x^1, s^1) \geq f^*$ due to (46). Now assume that $h_{k-1} - f^* \geq 0$. Rewrite as

$$
\begin{aligned}
h_k &= \beta\big[\bar{f}_{k-1} + \langle d_{k-1}, x^k \rangle - \gamma_{k-1}\big] + (1 - \beta)\big[f(x^k, s_k) + \langle g_k, x^k \rangle - \langle g_k, x^k \rangle\big] \\
&= \beta\big[\bar{f}_{k-1} + \langle d_{k-1}, x^{k-1} \rangle - \gamma_{k-1} + \langle d_{k-1}, x^k - x^{k-1} \rangle\big] + (1 - \beta)f(x^k, s_k) \\
&= \beta h_{k-1} + \beta \langle d_{k-1}, x^k - x^{k-1} \rangle + (1 - \beta)f(x^k, s_k).
\end{aligned}
$$

Using the update rule $x^k = x^{k-1} - \tau_{k-1} d_{k-1}$ in the above gives

$$
h_k = \beta(h_{k-1} - \tau_{k-1}\|d_{k-1}\|^2) + (1 - \beta)f(x^k, s_k). \tag{47}
$$

Recall that $\tau_k = \frac{(h_k - f_*^k)_+}{\|d_k\|^2}$ due to $\alpha_k = +\infty$. Hence,

$$
\tau_{k-1}\|d_{k-1}\|^2 = (h_{k-1} - f_*^{k-1})_+ = (h_{k-1} - f^*)_+ = h_{k-1} - f^*
$$

where the last equality is the induction hypothesis. Re-arranging the above we get

$$
h_{k-1} - \tau_{k-1}\|d_{k-1}\|^2 = f^*. \tag{48}
$$

Plugging this equality into (47) gives

$$
h_k = \beta f^* + (1 - \beta)f(x^k, s_k) \geq f^*,
$$

due to $\beta \in [0, 1)$ and $f(x^k, s_k) \geq f^*$. This completes the induction, and we have further shown that

$$
h_k - f^* = (1 - \beta)\big(f(x^k, s_k) - f^*\big). \tag{49}
$$

**Step 2.** Due to (46) and $\rho_k = 1$, it holds $\bar{f}_k^* = f^* = f_*^k$. Hence, the assumptions of Lemma 4.1 are satisfied and we can apply (18), which implies in particular that the iterates $(x^k)$ are almost surely contained in the bounded set $B$. By assumption, we conclude that $\mathbb{E}\big[\|g_j\|^2 \mid \mathcal{F}_k\big] \leq G^2$ for all $j \leq k$. Using Jensen for the discrete probability measure induced by $\rho_{j,k}$, we have

$$
\|d_k\|^2 = \|\sum_{j=1}^{k} \rho_{j,k} g_j\|^2 \leq \sum_{j=1}^{k} \rho_{j,k}\|g_j\|^2.
$$

Thus, we conclude for the conditional expectation that $\mathbb{E}\big[\|d_k\|^2 \mid \mathcal{F}_k\big] \leq G^2$. By Step 1, we have $\tau_k = \frac{h_k - f^*}{\|d_k\|^2}$. We will use next that $(x, y) \mapsto x^2/y$ is convex for $x \in \mathbb{R}, y > 0$. From (43) and applying conditional expectation, we have

$$
\begin{aligned}
\mathbb{E}\big[\|x^{k+1} - x^*\|^2 \mid \mathcal{F}_k\big] &\leq \|x^k - x^*\|^2 - \mathbb{E}\left[\frac{(h_k - f^*)^2}{\|d_k\|^2} \mid \mathcal{F}_k\right] \\
&\leq \|x^k - x^*\|^2 - \frac{\mathbb{E}[h_k - f^* \mid \mathcal{F}_k]^2}{\mathbb{E}[\|d_k\|^2 \mid \mathcal{F}_k]} \\
&\overset{(49)}{=} \|x^k - x^*\|^2 - \frac{(1 - \beta)^2\mathbb{E}\big[f(x^k, s_k) - f^* \mid \mathcal{F}_k\big]^2}{\mathbb{E}[\|d_k\|^2 \mid \mathcal{F}_k]} \\
&\leq \|x^k - x^*\|^2 - \frac{(1 - \beta)^2(f(x^k) - f^*)^2}{G^2}.
\end{aligned}
$$

**Step 3.** Taking full expectation, using the law of total expectation, suming over $k = 1, \ldots, K$, dividing by $K$ and re-arranging gives

$$
\frac{1}{K}\sum_{k=1}^{K}\mathbb{E}\big[(f(x^k) - f^*)^2\big] \leq \frac{G^2\|x^1 - x^*\|^2}{K(1 - \beta)^2}. \tag{50}
$$

Now, due to Jensen's inequality we have $\mathbb{E}\left[(f(x^k) - f^*)^2\right] \geq \mathbb{E}\left[f(x^k) - f^*\right]^2$ and because the square-root is concave, it holds

$$\frac{1}{K}\sum_{k=1}^{K}\mathbb{E}\left[f(x^k) - f^*\right] \leq \sqrt{\frac{1}{K}\sum_{k=1}^{K}\mathbb{E}\left[f(x^k) - f^*\right]^2}.$$

Using the above together with (50), we obtain

$$\min_{k=1,\ldots,K}\mathbb{E}\left[f(x^k) - f^*\right] \leq \frac{1}{K}\sum_{k=1}^{K}\mathbb{E}\left[f(x^k) - f^*\right] \leq \frac{G\|x^1 - x^*\|}{\sqrt{K}(1 - \beta)}.$$

$\square$

The above result is basically identical to (Loizou et al., 2021, Thm. C.1), but also allowing for momentum. We make two remarks: the best constant is clearly achieved by $\beta = 0$, i.e. no momentum. While empirically, momentum helps in most cases, we can not show a theoretical improvement at this time. Second, we do not need to assume bounded gradient norms as done in (Loizou et al., 2021), because this follows from the descent property Lemma 4.1. However, this improvement could be achieved analogously for the the proof of (Loizou et al., 2021) based on our techniques.

