# OpenReview forum: "MoMo: Momentum Models for Adaptive Learning Rates"
_ICLR.cc/2024/Conference — ICLR 2024 Conference Withdrawn Submission_

### Official Review · Reviewer_wwBz · 2023-11-01

**Soundness:** 1 poor
**Presentation:** 2 fair
**Contribution:** 2 fair
**Rating:** 3
**Confidence:** 4

**Summary:**

This paper presents MoMo, a model-based adaptive learning rate method that reinterprets the standard min f(x) problem by constructing a model of the actual function as opposed to the batchwise functions that are typically used. The paper focuses on SGD with momentum (SGD-M) and Adam, showing new theoretical adaptive learning rates for each. Experiments are conducted to compare against SGD-M and Adam.

**Strengths:**

* The paper is well-written, and flows well. Overall, conclusions are grounded in theorems that are proven in the appendix.
* Experiments are shown across a wide range of architectures.

**Weaknesses:**

* Critically, this paper does not compare against other adaptive learning rate methods such as AdaBound [1], AdaShift [2], AdaDB [3], LipschitzLR [4], AdaDelta [5], or WNGrad [6]. As such, its actual performance compared to other state-of-the-art algorithms is unknown, since the above-listed algorithms all show that adaptive learning rates outperform standard SGD-M and Adam.
* From Figures 2-3, it seems that MoMo tends to use small adaptive learning rates. This is in contrast to other methods such as AdaDelta (see [5] Fig. 2) or LipschitzLR (see [4] Fig. 3). Does this mean that in comparison, MoMo takes more iterations to reach an epsilon-ball of the minima? If so, does this mean MoMo is more computationally intensive?
* Although the experiments were repeated 3-5 times, it does not appear that any statistical tests were used to verify the significance of the improvements.

[1] Luo, Liangchen, et al. "Adaptive gradient methods with dynamic bound of learning rate." *arXiv preprint arXiv:1902.09843* (2019).

[2] Zhou, Zhiming, et al. "Adashift: Decorrelation and convergence of adaptive learning rate methods." *arXiv preprint arXiv:1810.00143* (2018).

[3] Yang, Liu, and Deng Cai. "AdaDB: An adaptive gradient method with data-dependent bound." *Neurocomputing* 419 (2021): 183-189.

[4] Yedida, Rahul, Snehanshu Saha, and  Tejas Prashanth. "Lipschitzlr: Using theoretically computed adaptive  learning rates for fast convergence." *Applied Intelligence* 51 (2021): 1460-1478.

[5] Zeiler, Matthew D. "Adadelta: an adaptive learning rate method." *arXiv preprint arXiv:1212.5701* (2012).

[6] Wu, Xiaoxia, Rachel Ward, and Léon Bottou. "Wngrad: Learn the learning rate in gradient descent." *arXiv preprint arXiv:1803.02865* (2018).

**Questions:**

* In Algorithm 3, what is the motivation for the new value of $f_*^k$? As a follow-up, from Appendix E.4, it seems the ResetStar and EstimateStar are rather important to the overall algorithm. Are there ways to modify the base MoMo algorithm so this check is not needed? Alternatively, can the estimation be improved?
* In Algorithms 3 and 4, why in the $\max$, do you use $f_*^1$?

---

### Official Review · Reviewer_ZNHx · 2023-11-01

**Soundness:** 2 fair
**Presentation:** 3 good
**Contribution:** 2 fair
**Rating:** 3
**Confidence:** 3

**Summary:**

This paper discusses an approach to adapting learning rates automatically via forming a proximal point method that forms a model of the loss in the deep learning settings. The proposed update is computationally and memory efficient (same as SGD+momentum). The algorithm MoMo works both for the classic SGD-M and for adaptive methods such as Adam. They show results demonstrating robustness to the choice of learning rate across different deep learning datasets.

**Strengths:**

- Proposed algorithm is straightforward to implement and has minimal overhead. The problem of learning rate scheduling is well-motivated and contexualized within existing work.
- MoMo is justified from a model-based perspective and can be viewed an extension of the Polyak step size that incorporates momentum. The authors derive the update when a lower bound is known for the loss and also show how it can be used for adaptive optimizers.
- The paper is well written and clear to follow.
- Results with MoMo outperform the base versions of the optimizers with a fixed learning rate.

**Weaknesses:**

My main concerns are with the empirical experiments. I do not believe it is a fair comparison to use a constant learning rate schedule for the baselines, when it is well-known that a learning rate schedule with decay generally improves performance (and is the schedule MoMo discovers, as shown in the appendix). It is straightforward to perform training runs with linear learning rate decay. Waterfall decay schedules are also commonly used on CIFAR and other image classification tasks. It would make the paper significantly stronger if MoMo was shown to perform comparatively or better to these learning rate schedules. In addition, conducting an experiment in a different domain or at a larger scale would improve the significance and confidence in the results.

**Questions:**

- Equation 6 seems like quite a loose approximation since the parameters can move substantially.
- It is a bit misleading to report results on Imagenet32 as Imagenet in the abstract.

I'm happy to adjust my score if the quality of baselines in empirical experiments is improved.

---

### Official Review · Reviewer_NNGe · 2023-11-02

**Soundness:** 3 good
**Presentation:** 3 good
**Contribution:** 2 fair
**Rating:** 3
**Confidence:** 4

**Summary:**

The paper proposed a new algorithm with new adaptive learning rates.

**Strengths:**

The intuition of the algorithmic design is mostly clear.

**Weaknesses:**

**Most experiments are too simple and toy, not convincing.**  It would be better to see how the performance of Momo methods on larger-scale experiments such as imagenet (instead of imagenet32) or larger NLP tasks (e.g. LLM pretraining such as GPT2 on OpenwebText or the Pile datasets ), or diffusion models. Without any of these experiments, it is hard to judge the effectiveness of the proposed method.

**Questions:**

1. What is the practical advantage of Momo methods compared to SGD or Adam? Is there any training speed-up？ Does Momo methods bring better performance? How does it save the trouble of hyperparameter tuning?  All of these are unclear or buried deeply somewhere in the script.

2. As mentioned in Remark 2.3, Momo methods require additional O(d) operations in **each iteration**. Please compare the total wall-clock running time of Momo methods and that of Adam or SGD. I suspect the additional operations in Momo will become a major computational overhead especially when the model size grows.

3. There are too many versions of the proposed method, which can be overwhelming, and it's challenging to decide which one to choose. Please provide a simple and clear user guide. If I'm working on an NLP task, which version should I use, and what hyperparameters should I choose? If it's a CV task, which version should I use, and what hyperparameters should I choose? Especially, can the authors summarize the total hyperparameters? Right now there are too many notations and too many versions of Momo. It is hard to tell which are tunable hyperparam and which are not. It would be better if there is a clean summary.